# Therapeutic paradigm of dual targeting VEGF and PDGF for effectively treating FGF-2 off-target tumors

Kayoko Hosaka[1], Yunlong Yang[1,2], Takahiro Seki [1,3], Qiqiao Du[1], Xu Jing[1,4], Xingkang He [1], Jieyu Wu [1], Yin Zhang[1,5], Hiromasa Morikawa[6], Masaki Nakamura[1,7], Martin Scherzer[1], Xiaoting Sun[1,8], Yuanfu Xu[9], Tao Cheng[9], Xuri Li[10], Xialin Liu[10], Qi Li[8], Yizhi Liu[10], An Hong[11], Yuguo Chen[12] & Yihai Cao[1✉]

FGF-2 displays multifarious functions in regulation of angiogenesis and vascular remodeling. However, effective drugs for treating FGF-2$^+$ tumors are unavailable. Here we show that FGF-2 modulates tumor vessels by recruiting NG2$^+$ pricytes onto tumor microvessels through a PDGFRβ-dependent mechanism. FGF-2$^+$ tumors are intrinsically resistant to clinically available drugs targeting VEGF and PDGF. Surprisingly, dual targeting the VEGF and PDGF signaling produces a superior antitumor effect in FGF-2$^+$ breast cancer and fibrosarcoma models. Mechanistically, inhibition of PDGFRβ ablates FGF-2-recruited perivascular coverage, exposing anti-VEGF agents to inhibit vascular sprouting. These findings show that the off-target FGF-2 is a resistant biomarker for anti-VEGF and anti-PDGF monotherapy, but a highly beneficial marker for combination therapy. Our data shed light on mechanistic interactions between various angiogenic and remodeling factors in tumor neovascularization. Optimization of antiangiogenic drugs with different principles could produce therapeutic benefits for treating their resistant off-target cancers.

[1] Department of Microbiology, Tumor and Cell Biology, Karolinska Institute, 171 77 Stockholm, Sweden. [2] Department of Cellular and Genetic Medicine, School of Basic Medical Sciences, Fudan University, Shanghai 200032, China. [3] Kagoshima University Graduate School of Medical and Dental Sciences, 890-8520 Kagoshima, Japan. [4] Department of Clinical Laboratory, The Second Hospital of Shandong University, Jinan 250033, China. [5] Medicine and Pharmacy Research Center, Binzhou Medical University, Yantai 264003 Shandong, China. [6] Unit of Computational Medicine, Department of Medicine, Center for Molecular Medicine, Karolinska Institute, 171 76 Stockholm, Sweden. [7] Department of Urology, The University of Tokyo Hospital 7-3-1, Hongo, Bunkyo-ku, 113-8655 Tokyo, Japan. [8] Department of Medical Oncology and Cancer Institute, Shuguang Hospital Shanghai University of Traditional Chinese Medicine, Shanghai 201203, China. [9] The State Key Laboratory of Experimental Hematology, Institute of Hematology and Blood Diseases Hospital, Chinese Academy of Medical Sciences and Peking Union Medical College, 288 Nanjing Road, Tianjin 300020, China. [10] State Key Laboratory of Ophthalmology, Zhongshan Ophthalmic Center, Sun Yat-Sen University, Guangzhou 510060 Guangdong, China. [11] Department of Cell Biology, Institute of Biomedicine, Jinan University, Guangzhou 510632, China. [12] Department of Emergency Medicine, Shandong Provincial Clinical Research Center for Emergency and Critical Care Medicine, Institute of Emergency and Critical Care Medicine of Shandong University, Qilu Hospital of Shandong University, Jinan 250012 Shandong, China. ✉email: yihai.cao@ki.se

In the tumor microenvironment (TME), malignant cells and stromal cells produce various angiogenic factors to induce neovascularization, vascular remodeling, tumor growth, and metastasis[1]. Targeting these factors and their signaling pathways provides attractive approaches for treating various cancers[1–4]. Indeed, antiangiogenic drugs (AADs) by blocking tumor-derived factors demonstrate clinical benefits in human cancer patients[3,5]. For examples, anti-vascular endothelial growth factor (VEGF) and anti-VEGF receptor (VEGFR) agents are commonly used drugs in the clinic for treating various cancers[6,7]. As the growth of all solid tumors depends on angiogenesis, targeting the tumor vasculature provides a generic approach for treating different types of cancers[8].

Tumor microvessels distinguish from the healthy ones by possessing several unique features, including disorganization, tortuosity, leakiness, low perfusion, instability, lacking sufficient perivascular cell coverage, and lacking completeness of the basement membrane[9,10]. These unique features represent imbalanced production of angiogenic factors, accumulation of metabolites, hypoxia, and alteration of the TME. VEGF is one of the key angiogenic factors that largely contribute to development of disorganized and highly leaky tumor vessels[11,12]. Treatment of tumors with VEGF blockades including neutralizing antibodies and its soluble receptors could restore a healthy vascular phenotype[10,13,14]. Currently, most clinically available AADs contain inhibitory components that target VEGF, VEGFR, and their downstream signaling[15]. The VEGF-VEGFR2 signaling is crucial for vascular patterning by the formation of endothelial cell tips at the leading edge of the growth cone[16].

In addition to VEGF, fibroblast growth factor 2 (FGF-2) and platelet-derived growth factors (PDGFs) significantly contribute to tumor neovascularization and vascular remodeling[17–21]. FGF-2 directly acts on endothelial cells through FGF receptors (FGFRs) to stimulate proliferation and to display a potent angiogenic effect[22–24]. It appears that FGF-2 and VEGF have distinguished biological activities on endothelial cells. While FGF-2 stimulates endothelial proliferation, VEGF mainly induces endothelial migration and tip formation[25]. PDGF-B is a main vascular remodeling factor acting on perivascular cells (PVCs), including pericytes and vascular smooth muscle cells[26]. Angiogenic endothelial cells produce PDGF-B to recruit PVCs onto the newly formed nascent vasculature[27]. Inhibition of the PDGF-B-PDGF receptor β (PDGFRβ) signaling ablates pericyte coverage of blood vessels in healthy and pathological tissues[28].

One of the key unsolved obstacles for antiangiogenic cancer therapy is the development of drug resistance[5]. Both intrinsic and evasive drug resistance counteract therapeutic benefits and compensatory production of factors that are not within the targets of AADs are recognized as a common mechanism[29,30]. The other key unsettled issue is to identify reliable biomarkers to predict therapeutic efficacy in cancer patients[5]. At this time of writing, such reliable biomarkers are not clinically available.

Our present study provides compelling evidence to support a therapeutic paradigm of therapeutic benefits by blocking angiogenic factors that are not targets of AADs. Surprisingly, combination of two off-targeted drugs produces superior anticancer effects, whereas either monotherapy has no effect on tumor growth. These astounding findings provide a concept and rationale for improving therapeutic efficacy of AADs by combination therapy.

## Results

### Dual inhibition of VEGF and PDGF limits FGF-2+ tumor growth.
To investigate the role of FGF-2 in angiogenesis, vascular remodeling, and tumor growth, we generated a murine mammary tumor model (E0771-FGF-2 tumor) that expressed a secretory form of FGF-2. Quantitative ELISA assay of tumor tissue samples showed that FGF-2 expressional level was 392 ng g$^{-1}$ tissue (Fig. 1a). Expression of FGF-2 did not significantly alter cell proliferation in vitro and VEGF and PDGF-B expression (Supplementary Fig. 1a, c, d). Interestingly, expression of FGF-2 in this cell line markedly accelerated the in vivo tumor growth rate (Fig. 1b). Systemic treatment of E0771 tumors with an anti-mouse VEGF neutralizing antibody (VEGF blockade) significantly inhibited tumor growth (64% inhibition) (Fig. 1c). However, E0771-FGF-2 tumors gained resistance toward anti-VEGF treatment and no statistical significance existed between the anti-VEGF-treated and non-treated groups (Fig. 1c). These results demonstrate that FGF-2 contributes to anti-VEGF drug resistance in a breast cancer model.

We next tested imatinib that primarily targets the PDGFR signaling, which was approved for treating chronic myeloid leukemia by targeting BCR/ABL and treating gastrointestinal stromal tumor. Imatinib monotherapy slightly suppressed tumor growth (42% inhibition) (Fig. 1d). Again, FGF-2 expression neutralized the antitumor effect of imatinib in this cancer model (Fig. 1d). In the E0771 tumor, a combination of VEGF blockade and imatinib produced an additive antitumor effect (78% inhibition) (Fig. 1e). Surprisingly, the same combination therapy also produced a similar antitumor effect (80% inhibition) in anti-VEGF or imatinib monotherapy-resistant E0771-FGF-2 tumors (Fig. 1e). These were unexpected findings because neither drug monotherapy significantly inhibited FGF-2+ tumor growth. We should emphasize while anti-VEGF had no impact on E0771 cancer cell proliferation in vitro, anti-PDGFRβ modestly inhibited tumor cell proliferation (Supplementary Fig. 1e, g).

Consistent with the antitumor effect, VEGF blockade significantly inhibited tumor angiogenesis in E0771 tumors (Fig. 1f, g). Imatinib monotherapy also significantly suppressed tumor neovascularization (Fig. 1f, h). Expectedly, E0771-FGF-2 tumors became antiangiogenic resistant in response to anti-VEGF monotherapy since FGF-2 also significantly augmented tumor angiogenesis and compromised the anti-VEGF sensitivity (Fig. 1f, g). The anti-VEGF and imatinib combination therapy further increased the antiangiogenic effect relative to their monotherapeutic regimens (Fig. 1f, i). Surprisingly, imatinib monotherapy further accelerated angiogenesis in E0771-FGF-2 tumors (Fig. 1f, h). Unexpectedly, the combination therapy ablated a majority of tumor microvessels in monotherapy-resistant E0771-FGF-2 tumors (Fig. 1f, i). In E0771 tumors, anti-VEGF treatment significantly increased the percentage of pericyte coverage in tumor microvessels, whereas imatinib ablated pericyte association with tumor vessels (Fig. 1f–h). In E0771-FGF-2 tumors, except imatinib significantly ablated perivascular cell coverage, anti-VEGF treatment either alone or in combination with imatinib had no impact on pericyte coverage (Fig. 1g–i). These results show that the anti-VEGF and imatinib combination therapy converts the monotherapy-resistant FGF-2+ tumors into highly sensitive tumors by synergistically targeting tumor angiogenesis.

### Vascular perfusion and hypoxia.
To study the functional impact of tumor vasculatures in response to various monotherapy and combination therapy, we measured blood perfusion and vascular permeability using lysinated Rhodamine-labeled 2000 kDa and 70 kDa dextrans[31,32]. While VEGF blockade reduced vascular perfusion in control tumors, it had no impact on E0771-FGF-2 tumors (Fig. 2a, c). A similar effect was also seen with imatinib monotherapy (Fig. 2a, c). Interestingly, anti-VEGF and imatinib combination therapy markedly inhibited blood perfusion in the E0771-FGF-2 tumors (Fig. 2a, c). These functional findings

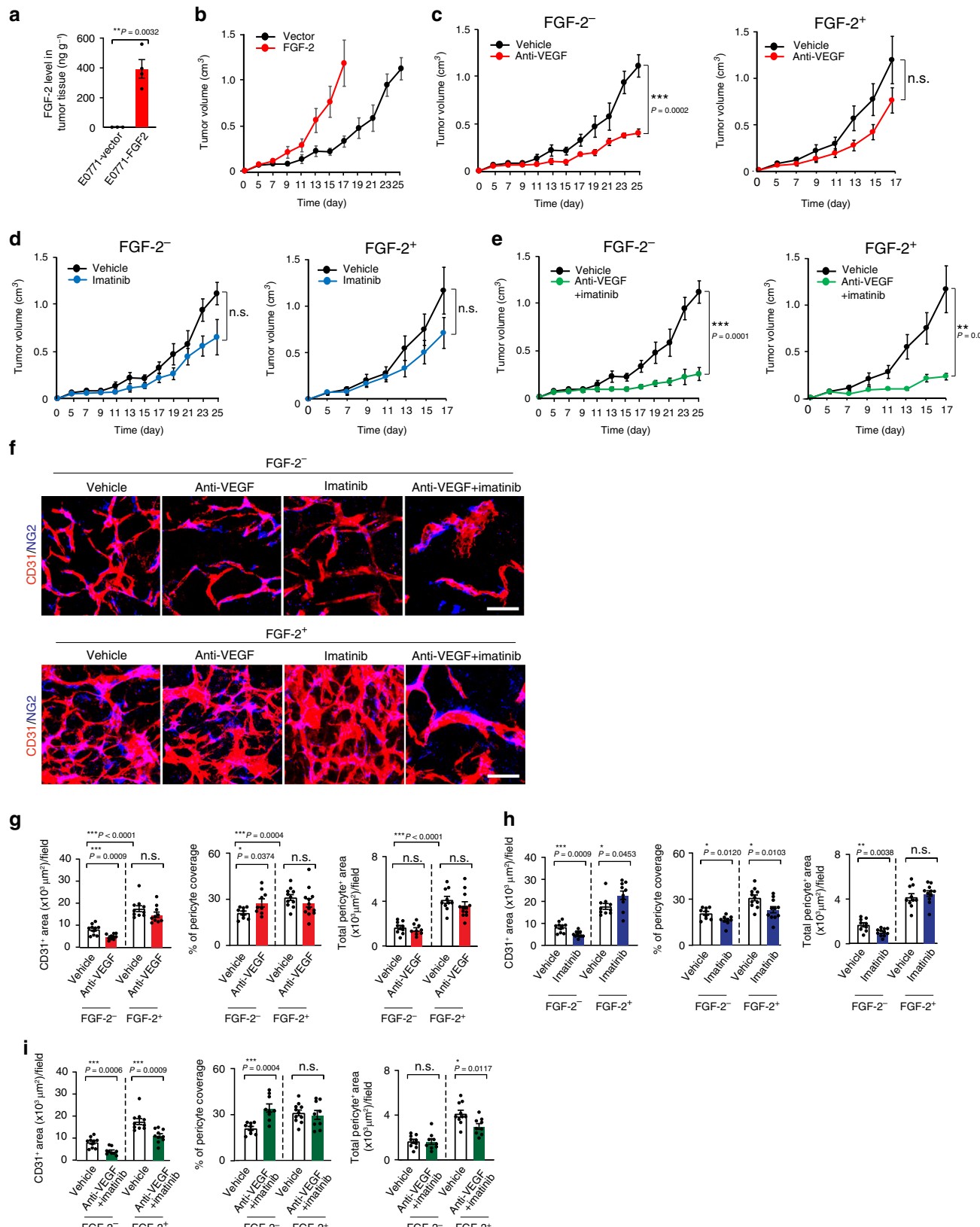

reconciled with the antiangiogenic effects of combination therapy. Consistent with previously published findings, anti-VEGF alone inhibited vascular leakage in control tumors (Fig. 2b, d). Similarly, anti-VEGF monotherapy also displayed a potent anti-permeability effect in E0771-FGF-2 tumors (Fig. 2b, d). Treatment of control and E0771-FGF-2 tumors with imatinib

monotherapy significantly altered vascular permeability (Fig. 2b, d). However, anti-VEGF and imatinib combination produced an additive effect against vascular leakage (Fig. 2b, d).

We next studied oxygen contents in mono- and combination-drug-treated control and FGF-2+ tumors. While anti-VEGF monotherapy increased hypoxia in control tumors, it had no

**Fig. 1 Growth rates and angiogenesis in various drug-treated FGF-2$^+$ and control breast cancers. a** ELISA measurement of FGF-2 levels in E0771-vector ($n = 3$) and E0771-FGF-2 tumor tissues ($n = 4$). $P = 0.0032$. **b** Tumor growth of E0771-vector and E0771-FGF-2 ($n = 5, 6$). **c** Tumor growth of vehicle- and anti-VEGF-treated E0771-vector ($n = 5, 6$; $P = 0.0002$). Tumor growth rates of vehicle- and anti-VEGF-treated E0771-FGF-2 ($n = 6$). **d** Tumor growth of vehicle- and imatinib-treated E0771-vector ($n = 5$). Tumor growth rates of vehicle- and imatinib-treated E0771-FGF-2 ($n = 6$). **e** Tumor growth of vehicle- and anti-VEGF plus imatinib-treated E0771-vector ($n = 5, 6$; $P = 0.0001$). Tumor growth rates of vehicle- and anti-VEGF plus imatinib-treated E0771-FGF-2 ($n = 6, 8$; $P = 0.0010$). **f** CD31$^+$ microvessels (red) and NG2$^+$ pericytes (blue) in various drug-treated E0771-vector and E0771-FGF-2 cancers. Bar = 50 μm. **g** Quantification of microvessels ($n = 10$ each; $P$(Vehicle-treated-vector vs anti-VEGF-treated-vector) = 0.0009), pericyte coverage ($n = 9/9/11/12$; $P$(Vehicle-treated-vector vs anti-VEGF-treated-vector) = 0.0374) and pericyte area ($n = 9/9/10/12$) of vehicle- and anti-VEGF-treated E0771 vector and E0771-FGF-2. **h** Quantification of microvessels ($n = 10/11/10/10$; $P$(Vehicle-treated-vector vs imatinib-treated-vector) = 0.0009; $P$(Vehicle-treated-FGF-2 vs anti-VEGF-treated-FGF-2) = 0.0453), pericyte coverage ($n = 9/9/11/11$; $P$(Vehicle-treated-vector vs imatinib-treated-vector) = 0.0120; $P$(Vehicle-treated-FGF-2 vs imatinib-treated-FGF-2) = 0.0103) and pericyte area ($n = 9/11/10/11$; $P$(Vehicle-treated-vector vs imatinib-treated-vector) = 0.0038) of vehicle- and imatinib-treated E0771-vector and E0771-FGF-2. **i** Quantification of microvessels ($n = 10/9/10/10$; $P$(Vehicle-treated-vector vs combination-treated-vector) = 0.0006; $P$(Vehicle-treated-FGF-2 vs combination therapy-treated-FGF-2) = 0.0009), pericyte coverage ($n = 9/9/11/9$; $P$(Vehicle-treated-vector vs combination-treated-vector) = 0.0004) and pericyte area ($n = 9/9/10/9$; $P$(Vehicle-treated-FGF-2 vs combination-treated-FGF-2) = 0.0117) of vehicle- and anti-VEGF plus imatinib-treated E0771-vector and E0771-FGF-2 cancers. FGF-2$^-$ = vector cancers; FGF-2$^+$ = FGF-2 cancers; n.s. Not significant; $*P < 0.05$; $**P < 0.01$; $***P < 0.001$; two-tailed Student $t$-test. The exact P-values indicate in panels. **a–e**; $n$ indicates individual mice. Data presented as mean ± s.e.m. **g–i**; Data presented as mean from random images of 4 animals/group ± s.e.m. Experiments were repeated twice. Source data are provided as a Source Data file.

effect in E0771-FGF-2 tumors (Fig. 2e, f). Similarly, imatinib alone increased tumor hypoxia in control tumors but no effect on FGF-2$^+$ tumors (Fig. 2e, f). In contrast, anti-VEGF and imatinib combination elevated hypoxia in both tumor types. These findings reconcile with the inhibitory effects of angiogenesis and vascular functions by these drugs.

**Additive anti-proliferative effect by combination therapy.** Anti-VEGF monotherapy produced a significant effect of suppressing tumor cell proliferation (Fig. 3a, b) in control E0771 tumors. Imatinib alone also significantly inhibited tumor cell proliferation. Combination therapy produced a more pronounced anti-proliferative effect than their respective monotherapy (Fig. 3a, b). Neither VEGF blockage nor imatinib had any impact on tumor cell proliferation in E0771-FGF-2 tumors. However, anti-VEGF and imatinib combination significantly suppressed tumor cell proliferation in E0771-FGF-2 tumors (Fig. 3a, b). Furthermore, the number of CD31$^+$ and Ki67$^+$ double-positive cells was markedly decreased in the combination-treated tumors (Fig. 3c), indicating inhibition of proliferating endothelial cells as a part of the mechanism of antitumor activity. These data show that anti-VEGF and imatinib synergistically inhibited tumor cell proliferation in FGF-2$^+$ tumors.

Conversely, anti-VEGF alone induced cellular apoptosis in control tumors, but had no effect on FGF-2$^+$ tumors (Fig. 3d). A similar apoptotic pattern had also been seen with imatinib monotherapy. Despite indolent apoptotic effects of their monotherapy, combination therapy with VEGF blockade plus imatinib yielded a apoptotic effect in FGF-2$^+$ tumors (Fig. 3d, e). In FGF$^-$ control tumors, combination therapy by simultaneous blocking of VEGF and PDGFR signaling increased the necrotic area in tumors (Supplementary Fig. 2a). In contrast, the combination therapy had no effect on FGF-2$^+$ tumor necrosis (Supplementary Fig. 2b). These results demonstrate that anti-VEGF and imatinib combination therapy synergistically inhibits tumor cell proliferation and increases cellular apoptosis in FGF-2$^+$ tumors.

To exclude other mechanistic possibilities of combination therapy in suppressing tumor growth by altering the TME, other cellular components including inflammatory cells and fibroblasts were analyzed. In both FGF-2$^-$ and FGF-2$^+$ E0771 tumors, anti-VEGF or imatinib monotherapy and their combination therapy did not alter fibrotic components (Supplementary Fig. 2c–f). Immunohistochemistry and FACS analyses were performed to define specific immune cell populations. In FGF-2$^+$ E0771 breast cancers, a significant increase of Iba1$^+$ tumor-associated

macrophages (TAMs) was detected by immunostaining (Supplementary Fig. 3a–c). This finding was validated by FACS analysis of F4/80$^+$ TAM population (Supplementary Fig. 3g). However, anti-VEGF and imatinib monotherapy and anti-VEGF + imatinib therapy did not change the ratios of Iba1$^+$ and F4/80$^+$ TAMs (Supplementary Fig. 3a–c, g). Interestingly, both immunohistochemistry and FACS analyses showed that the CD3$^+$ total T cell population was decreased in FGF-2$^+$ E0771 tumors relative to FGF-2$^-$ control tumors (Supplementary Fig. 3a, b, d, h). Again, monotherapy and combination therapy did change the ratio of the total T cell population. In-depth analysis showed that CD4$^+$ T cell subpopulation was significantly decreased in the FGF-2$^+$ E0771 tumors (Supplementary Fig. 3a, b, e, i). Monotherapy and combination did not alter the ratio of this subpopulation of T cells (Supplementary Fig. 3a, b, e, i). By contrast, the CD8$^+$ T cell subpopulation remained unchanged in FGF-2$^-$ and FGF-2$^+$ tumors regardless of treatments (Supplementary Fig. 3a, b, f, j). These findings suggest to us that vascular suppression is likely the principal mechanism of antitumor activity.

**Synergistic anti-FGF-2$^+$ fibrosarcoma by dual inhibition.** To validate our findings from the breast cancer model, we carried out an independent study using a murine fibrosarcoma model. In this model, a secreted form of FGF-2 was expressed to an equivalent, but slightly higher, level of E0771 (Fig. 4a). Similar to E0771, FGF-2 expression did not significantly affect tumor cell growth rate in vitro and VEGF and PDGF-B expression (Supplementary Fig. 1b, c, d). The conditioned media from FGF-2$^+$ E0771 and T241 cells significantly stimulated proliferation of endothelial cell and pericytes in vitro, which was equivalent to the stimulatory effect of a recombinant FGF-2 protein (Supplementary Fig. 1i, j). These data demonstrated that FGF-2 released from tumor cells was biologically active.

Expectedly, the fibrosarcoma was also sensitive to anti-VEGF monotherapy and around 80% of tumor suppression was achieved after 3-week therapy (Fig. 4b). Imatinib monotherapy also significantly inhibited tumor growth in the FGF-2$^-$ fibrosarcoma (Fig. 4c). Again, neither VEGF blockade nor imatinib monotherapy produced any significant antitumor activity in FGF-2$^+$ fibrosarcomas (Fig. 4b, c), indicating that FGF-2 significantly contributes to development of drug resistance through a compensatory mechanism. Similar to the breast cancer model, combination of VEGF blockade and imatinib produced an astoundingly potent antitumor effect. The therapeutic efficacy of this combination regimen was indistinguishable between FGF-2$^-$

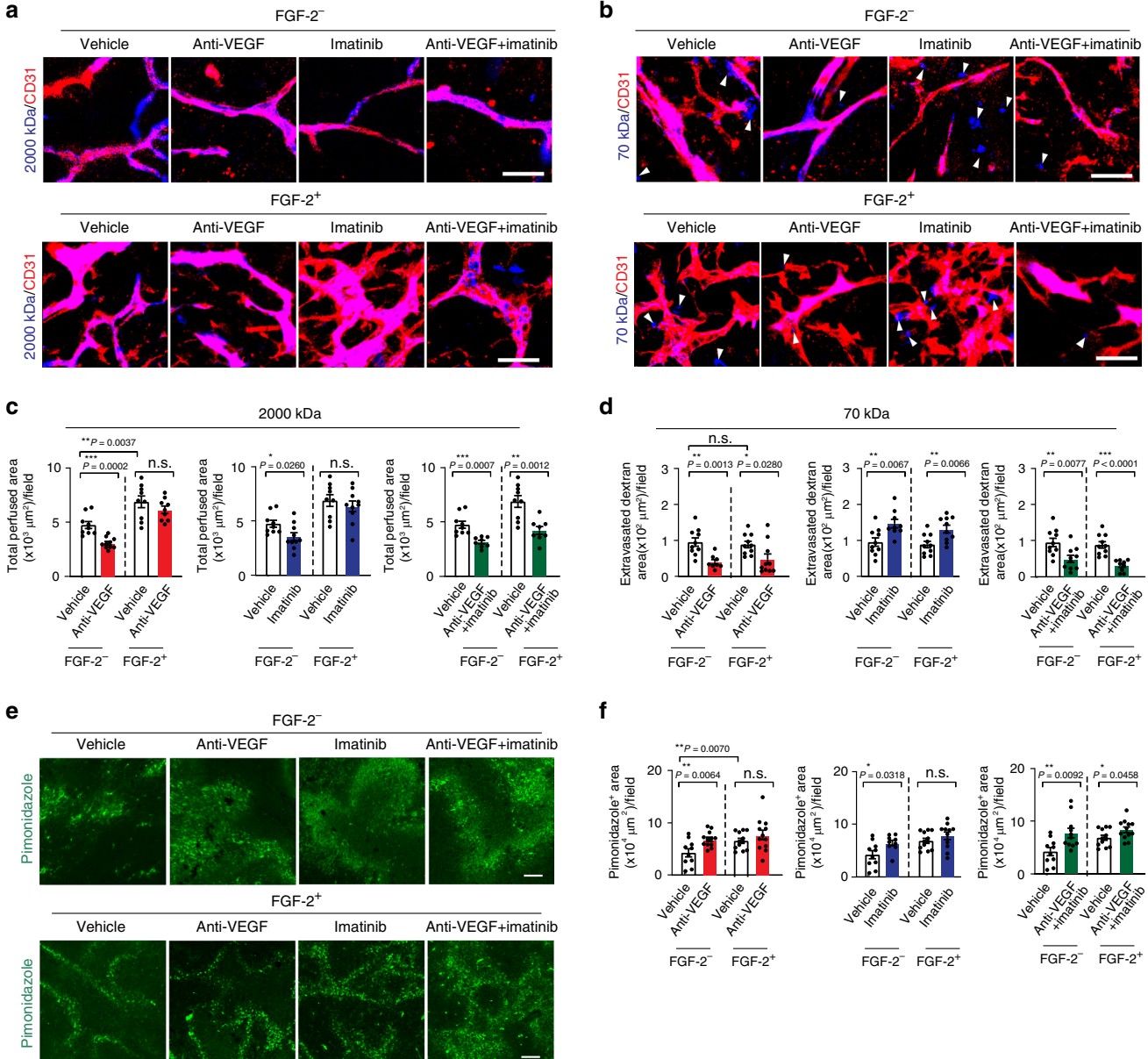

**Fig. 2 Vascular perfusion, vascular permeability, and tumor hypoxia. a** Vascular perfusion of Rhodamine-labeled lysinated 2000 kDa dextran (blue) of various monotherapy- and combination therapy-treated E0771-vector and E0771-FGF-2 breast cancers. Red indicates CD31+ microvessels. Bar = 50 μm. **b** Vascular permeability of Rhodamine-labeled lysinated 70 kDa dextran (blue) of various monotherapy- and combination therapy-treated E0771-vector and E0771-FGF-2 breast cancers. Red indicates CD31+ microvessels. Bar = 50 μm. Arrowheads indicate extravasation of 70 kDa dextran from the tumor vasculature. **c** Quantification of vascular perfusion of vehicle-, anti-VEGF-, imatinib- and combination therapy-treated E0771-vector and E0771-FGF-2 breast cancers (n(Vector) = 9/10/10/8; n(FGF-2) = 9/9/10/8; P(Vector vs FGF-2) = 0.0037; P(Vehicle-treated vector vs anti-VEGF-treated vector) = 0.0002; P(Vehicle-treated vector vs imatinib-treated vector) = 0.0260; P(Vehicle-treated vector vs combination therapy-treated vector) = 0.0007; P(Vehicle-treated FGF-2 vs combination therapy-treated FGF-2) = 0.0012. **d** Quantification of vascular permeability of vehicle-, anti-VEGF-, imatinib- and combination therapy-treated E0771-vector and E0771-FGF-2 breast cancers (n(Vector) = 10/8/9/10; n(FGF-2) = 10/10/10/9). P(Vehicle-treated vector vs anti-VEGF-treated vector) = 0.0013, P(Vehicle-treated FGF-2 vs anti-VEGF-treated FGF-2) = 0.0280, P(Vehicle-treated vector vs imatinib-treated vector) = 0.0067, P(Vehicle-treated FGF-2 vs imatinib-treated FGF-2) = 0.0066, P(Vehicle-treated vector vs combination therapy-treated vector) = 0.0077, P(Vehicle-treated FGF-2 vs combination therapy-treated FGF-2) < 0.0001. **e** Pimonidazole+ hypoxic signals (green) in various monotherapy- and combination therapy-treated E0771-vector and E0771-FGF-2 breast cancer tissues. Bar = 100 μm (**f**). Quantification of pimonidazole+ hypoxic signals of vehicle-, anti-VEGF-, imatinib- and combination therapy-treated E0771-vector and E0771-FGF-2 breast cancers (n(Vector) = 10/12/9/11; n(FGF-2) = 12 each; P(Vector vs FGF-2) = 0.0070; P(Vehicle-treated vector vs anti-VEGF-treated vector) = 0.0064; P(Vehicle-treated vector vs imatinib-treated vector) = 0.0318; P(Vehicle-treated vector vs combination therapy-treated vector) = 0.0092; P(Vehicle-treated FGF-2 vs combination therapy-treated FGF-2) = 0.0458). FGF-2− = vector cancers; FGF-2 = FGF-2 cancers; n.s. Not significant; *P < 0.05; **P < 0.01; ***P < 0.001; two-tailed student t-test. Data presented as mean from random fields ± s.e.m. Experiments were independently repeated twice. Source data are provided as a Source Data file.

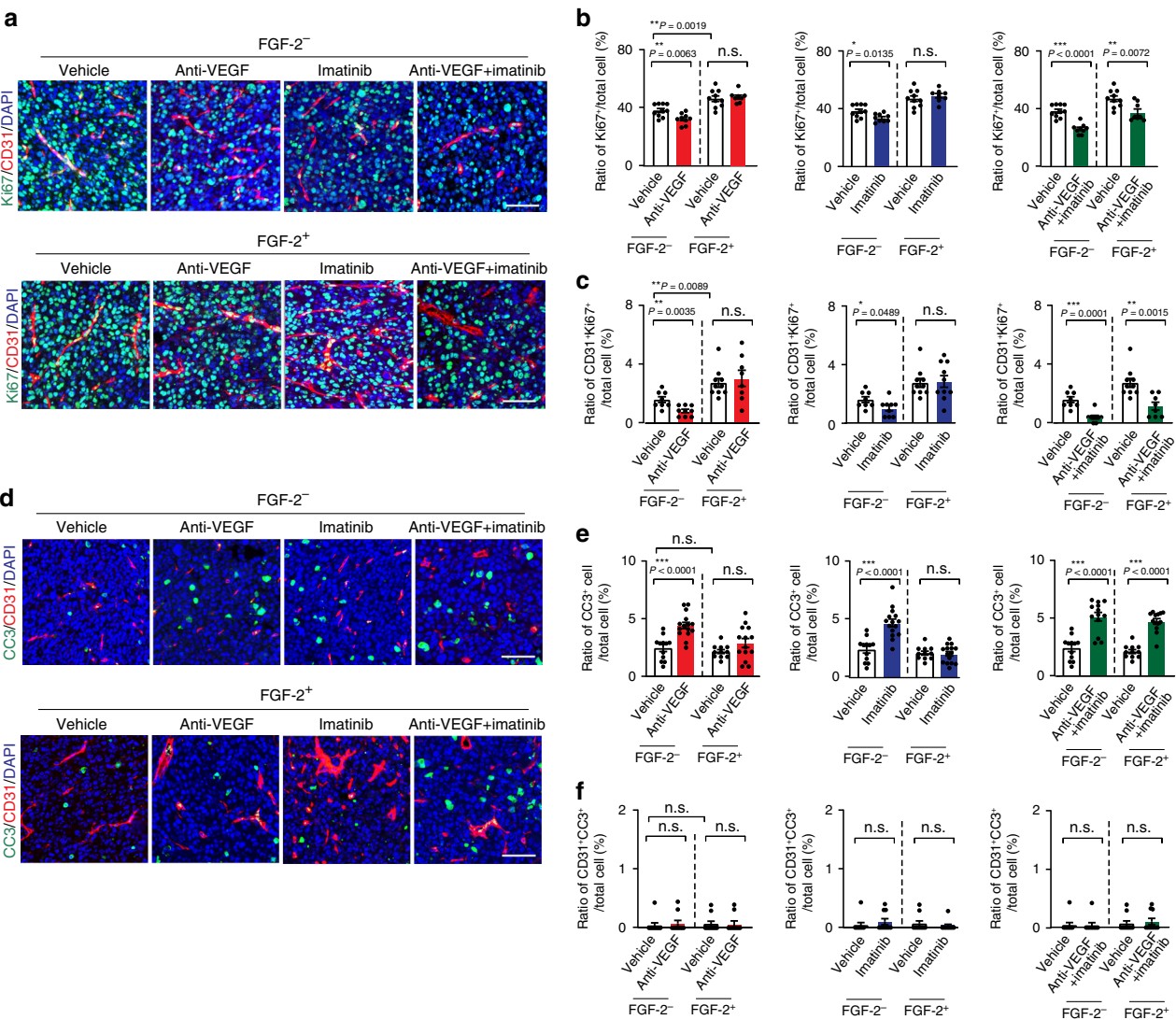

**Fig. 3 Tumor cell proliferation and apoptosis of FGF-2 breast cancers in response to various drug treatment. a** Ki67+ proliferative cell signals (green) co-stained with CD31+ microvessels (red) and DAPI (blue) of various monotherapy and combination therapy-treated E0771-vector and E0771-FGF-2 breast cancer tissues. Bar = 50 μm. **b** Quantification of Ki67+ signals in vehicle-, anti-VEGF-, imatinib- and combination therapy-treated E0771-vector and E0771-FGF-2 breast cancers (n(Vector) = 10/9/9/9; n(FGF-2) =10/8/8/8; P(Vector vs FGF-2) = 0.0019; P(Vehicle-treated vector vs anti-VEGF-treated vector) = 0.0063; P(Vehicle-treated vector vs imatinib-treated vector) = 0.0135; P(Vehicle-treated vector vs the combination therapy-treated vector) < 0.0001; P(Vehicle-treated FGF-2 vs the combination therapy-treated FGF-2) = 0.0072). **c** Quantification of Ki67+ and CD31+ double-positive signals in vehicle-, anti-VEGF-, imatinib- and combination therapy-treated E0771-vector and E0771-FGF-2 breast cancers (n(Vector) = 8/9/9/9; n(FGF-2) = 10/8/10/8; P(Vector vs FGF-2) = 0.0089; P(Vehicle-treated vector vs anti-VEGF-treated vector) = 0.0035; P(Vehicle-treated vector vs imatinib-treated vector) = 0.0489; P(Vehicle-treated vector vs the combination therapy-treated vector) = 0.0001; P(Vehicle-treated FGF-2 vs the combination therapy-treated FGF-2) = 0.0015). **d** Micrographs of caspase-3+ apoptotic tumor cells (green) co-stained with CD31+ microvessels (red) and DAPI (blue) in various monotherapy- and combination therapy-treated E0771-vector and E0771-FGF-2 breast cancers. Bar = 50 μm. **e** Quantification of caspase-3+ signals in vehicle-, anti-VEGF-, imatinib- and combination therapy-treated E0771-vector and E0771-FGF-2 breast cancers (n(Vector) = 12/15/15/13; n(FGF-2) =11/14/15/13;; P(Vehicle-treated vector vs anti-VEGF-treated vector) < 0.0001; P(Vehicle-treated vector vs imatinib-treated vector) < 0.0001; P(Vehicle-treated vector vs combination therapy-treated vector) < 0.0001; P(Vehicle-treated FGF-2 vs combination therapy-treated FGF-2) < 0.0001). **f** Quantification of caspase-3+ -CD31+ double-positive signals in vehicle-, anti-VEGF-, imatinib- and combination therapy-treated E0771-vector and E0771-FGF-2 breast cancers (n(Vector) = 10 each; n(FGF-2) = 10 each). FGF-2− = vector cancers; FGF-2+ = FGF-2 cancers; n.s. Not significant; *P < 0.05; **P < 0.01; ***P < 0.001; two-tailed student t-test. Data presented as mean from random fields ± s.e.m. Experiments were independently repeated twice. Source data are provided as a Source Data file.

and FGF-2+ fibrosarcomas (approximate 80% tumor suppression in both models) (Fig. 4d). Neither anti-VEGF nor anti-PDGFRβ had any effect on T241 cell proliferation in vitro (Supplementary Fig. 1f, h).

The synergistic antitumor activity of combination therapy is well correlated with the antiangiogenic activity in FGF-2+ fibrosacomas (Supplementary Fig. 4). Similar inhibitory effects

on blood perfusion and permeability by monotherapy and combination therapy were also seen in the fibrosarcoma model (Fig. 4e–h). Consequently, reduction of tumor vessel density and blood perfusion by combination therapy led to significant increases of tumor hypoxia, which would be otherwise insensitive to anti-VEGF and imatinib monotherapy in FGF-2+ fibrosarcomas (Fig. 4i, j). These independent findings validate the fact that

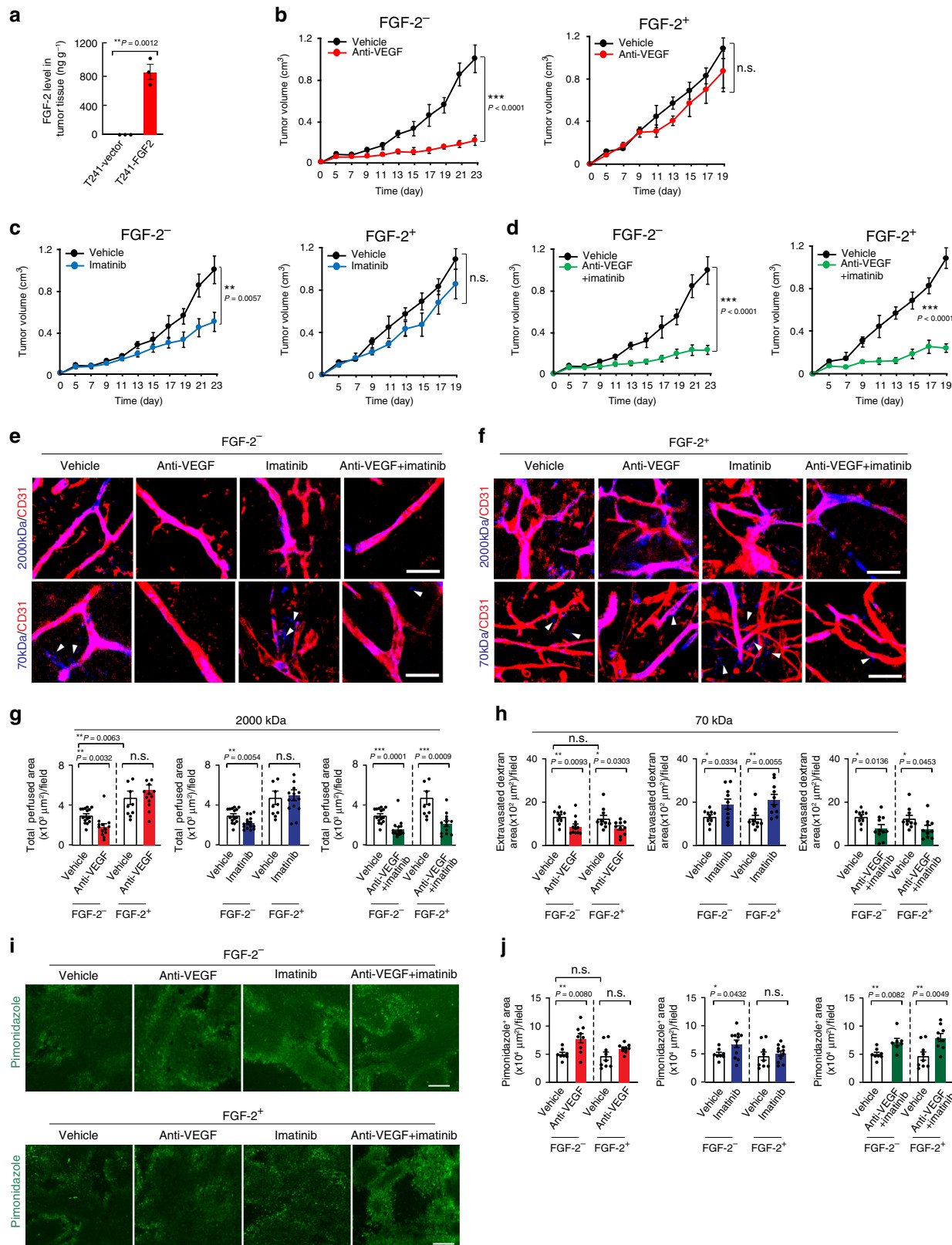

anti-VEGF and imatinib combination therapy is highly effective in suppressing FGF-2+ tumor growth and angiogenesis, which would be otherwise resistant to their monotherapy.

Consistent with the antitumor effect, anti-VEGF and imatinib combination effectively suppressed tumor cell proliferation in FGF-2− fibrosarcomas (Fig. 5a–c). In contrast, their monotherapy

had no impact on tumor cell proliferation in FGF-2+ tumors, although anti-VEGF and imatinib monotherapy significantly inhibited tumor cell proliferation in FGF-2− tumors (Fig. 5a–c). Similar to the breast cancer model, anti-VEGF and imatinib combination markedly induced cellular apoptosis in FGF-2+ fibrosarcomas, which otherwise were completely resistant in their

**Fig. 4 Tumor growth rates, vascular function, and hypoxia in various drug-treated FGF-2$^+$ and control fibrosarcomas. a** Expression levels of FGF-2 protein in T241-vector and T241-FGF-2 tumors ($n = 3$; $P$(Vector vs FGF-2) = 0.0012). **b** Tumor growth of vehicle- and anti-VEGF-treated T241-vector ($n = 7, 10$) and T241-FGF-2 ($n = 6$; $P$(Vehicle-treated-vector vs anti-VEGF-treated-vector) < 0.0001). **c** Tumor growth of vehicle- and imatinib-treated T241-vector ($n = 7, 10$; $P$(Vehicle-treated-vector vs imatinib-treated-vector) = 0.0057) and T241-FGF-2 ($n = 6, 7$). **d** Tumor growth of vehicle- and anti-VEGF plus imatinib-treated T241-vector ($n = 7, 10$; $P$(Vehicle-treated-vector vs combination-treated-vector) < 0.0001) and T241-FGF-2 ($n = 6, 7$; $P$(Vehicle-treated-FGF-2 vs combination-treated-FGF-2) < 0.0001). **e** Vascular perfusion of 2000 kDa dextran (blue) stained with CD31$^+$ microvessels (red) of various therapy-treated T241-vector and T241-FGF-2. Bar = 50 μm. **f** Vascular permeability of 70 kDa dextran (blue) stained with CD31$^+$ microvessels (red) of various therapy-treated T241-vector and T241-FGF-2. Arrowheads indicate the extravasated dextran. Bar = 50 μm. **g** Quantification of vascular perfusion of vehicle-, anti-VEGF-, imatinib- and combination therapy-treated T241-vector and T241-FGF-2 fibrosarcomas ($n$(Vector) = 15/13/15/15; n (FGF-2) = 10/15/15/13; $P$(Vehicle-treated-vector vs anti-VEGF-treated-vector) = 0.0032; $P$(Vehicle-treated-vector vs imatinib-treated-vector) = 0.0054; $P$(Vehicle-treated-vector vs combination-treated-vector) = 0.0001; $P$(Vehicle-treated-FGF-2 vs combination-treated-FGF-2) = 0.0009). **h** Quantification of vascular permeability of vehicle-, anti-VEGF-, imatinib-, and combination therapy-treated T241-vector and T241-FGF-2 ($n$(Vector) = 10/12/10/11; n (FGF-2) = 10 each; $P$(Vehicle-treated-vector vs anti-VEGF-treated-vector) = 0.0093; $P$(Vehicle-treated-FGF-2 vs anti-VEGF-treated-FGF-2) = 0.0303; $P$(Vehicle-treated-vector vs imatinib-treated-vector) = 0.0334; $P$(Vehicle-treated-FGF-2 vs imatinib-treated-FGF-2) = 0.0055; $P$(Vehicle-treated-vector vs combination-treated-vector) = 0.0136; $P$(Vehicle-treated-FGF-2 vs combination-treated-FGF-2) = 0.0453). **i** Pimonidazole$^+$ hypoxic signals (green) in various therapy-treated T241-vector and T241-FGF-2. Bar = 100 μm. **j** Quantification of pimonidazole$^+$ hypoxic signals of vehicle-, anti-VEGF-, imatinib- and combination therapy-treated T241-vector and T241-FGF-2. ($n$(Vector) = 8/9/12/7; n(FGF-2) = 9/9/11/9;). FGF-2$^-$ = vector cancers; FGF-2$^+$ = FGF-2 cancers; n.s. = Not significant; two-tailed $t$-test. The exact P-values indicate in a figure. **a–d**; $n$ indicates individual mice. Data presented as mean ± s.e.m. **g, h, j**; Data presented as mean from random fields of 4 animals/group ± s.e.m. Experiments were repeated twice. Source data are provided as a Source Data file.

monotherapy settings (Fig. 5d–f). Similar to the breast cancer model, anti-VEGF and imatinib combination therapy increased the necrotic area in FGF-2$^-$ tumors, whereas this therapeutic regiment had no effect on necrosis of FGF-2$^+$ tumors (Supplementary Fig. 5a, b). Furthermore, mono- or combination therapy did not alter the FSP1$^+$ and αSMA$^+$ fibrotic components in FGF-2$^-$ and FGF-2$^+$ tumors (Supplementary Fig. 5c–f). Similar to the E0771 breast cancer model, FGF-2 expression significantly increased recruitment of the Iba1$^+$ and F4/80$^+$ TAMs as detected by immunohistochemistry and FACS analyses (Supplementary Fig. 6a–c, g). However, anti-VEGF or imatinib monotherapy and anti-VEGF + imatinib combination therapy did not change the TAM populations relative to their respective controls (Supplementary Fig. 6). Also, the total CD3$^+$ population, CD4$^+$ subpopulation, and CD8$^+$ subpopulation of T cells remained unchanged in all untreated and treated groups (Supplementary Fig. 6). These data demonstrate that anti-VEGF and imatinib synergistically inhibit the growth of their monotherapy-resistant FGF-2 tumors by targeting the tumor vasculature.

In addition to this tumor model, we also studied the impact of anti-VEGF plus imatinib combination therapy on a well-established tumor. Similarly, the combination therapy produced a synergistic antitumor activity on FGF-2$^+$ fibrosarmas by inhibiting tumor angiogenesis (Supplementary Fig. 7).

**Inhibiting FGF-2$^+$ tumors by dual blocking VEGF and PDGFRβ.** Since imatinib targets a broad-spectrum of kinases, its synergistic antitumor effects with VEGF blockade might involve inhibition of multiple kinases other than PDGFR. To exclude this possibility, we employed an anti-PDGFRβ specific neutralizing monoclonal antibody in the monotherapy and combination therapeutic settings. FGF-2$^+$ tumors were highly resistant to anti-VEGF and anti-PDGFRβ monotherapy and no significant antitumor activity was detected (Fig. 6a). By contrast, combination of anti-VEGF and anti-PDGFRβ blockades produced very potent antitumor activity. These findings provide compelling evidence that simultaneous blocking VEGF- and PDGFRβ-triggered signaling pathways convert FGF-2$^+$ resistant tumors to sensitive tumors.

In the FGF-2$^+$ tumors, anti-VEGF treatment did not inhibit tumor angiogenesis (Fig. 6b, c). Unexpectedly, anti-PDGFRβ alone treatment augmented tumor angiogenesis in FGF-2$^+$

tumors. These findings support the notion from the imatinib-treated tumors that ablation of perivascular cells by PDGFR inhibitors allowed excessive sprouting of the tumor vasculatures. Consistent with their synergistic antitumor activity, combination of VEGF and PDGFRβ blockades markedly inhibited tumor angiogenesis, blood perfusion, leakiness (Fig. 6b–e). These findings validate the synergistic antitumor activity by combination of anti-VEGF agent with imatinib through the mechanism of suppressing tumor angiogenesis. Consequently, the combination-treated FGF-2$^+$ tumors experienced a high degree of hypoxia, marked repression of tumor cell proliferation, and significantly increased apoptosis (Fig. 6f–l). These results show that simultaneous targeting PDGFRβ- and VEGF-triggered signaling events are necessary and sufficient to circumvent FGF-2-induced compensatory resistance.

**Impacts of combination therapy on chemotherapy.** In clinical settings and for most cancer types, antiangiogenic therapy in combination with chemotherapy often produces clinical benefits. To study if our anti-VEGF and anti-PDGFR combination therapy would further enhance antitumor effects of chemotherapy, we treated breast cancer and fibrosarcoma by triple combination therapy, consisting of VEGF blockade, imatinib, and 5-Fluorouracil (5-FU). Two therapeutic doses of a low dose (10 mg kg$^{-1}$) and a high dose (60 mg kg$^{-1}$) of 5-FU were combined with VEGF blockade and imatinib. At 10 mg kg$^{-1}$, 5-FU did not significantly inhibit E0771 breast cancer growth and addition of 5-FU did not affected the growth rate of tumors treated with VEGF blockade plus imatinib (Supplementary Fig. 8a). At 60 mg kg$^{-1}$, 5-FU alone significantly inhibited tumor growth, which was nearly equivalent to the antitumor effect of VEGF blockade plus imatinib (Supplementary Fig. 8b). Triple combination of 5-FU + VEGF blockade + imatinib significantly improved antitumor activity (Supplementary Fig. 8b).

In the fibrosarcoma model, the low dose 10 mg kg$^{-1}$ of 5-FU markedly inhibited tumor growth, indicating that fibrosarcoma was hypersentitive to 5-FU relative to the breast cancer model (Supplementary Fig. 8c). Triple combination of 5-FU + VEGF blockade + imatinib further increased the antitumor activity (Supplementary Fig. 8c). Similarly, the triple combination therapeutic regimen at 60 mg kg$^{-1}$ also improved antitumor effect (Supplementary Fig. 8d). These findings indicate that combination of dual inhibition of VEGF and PDGF with

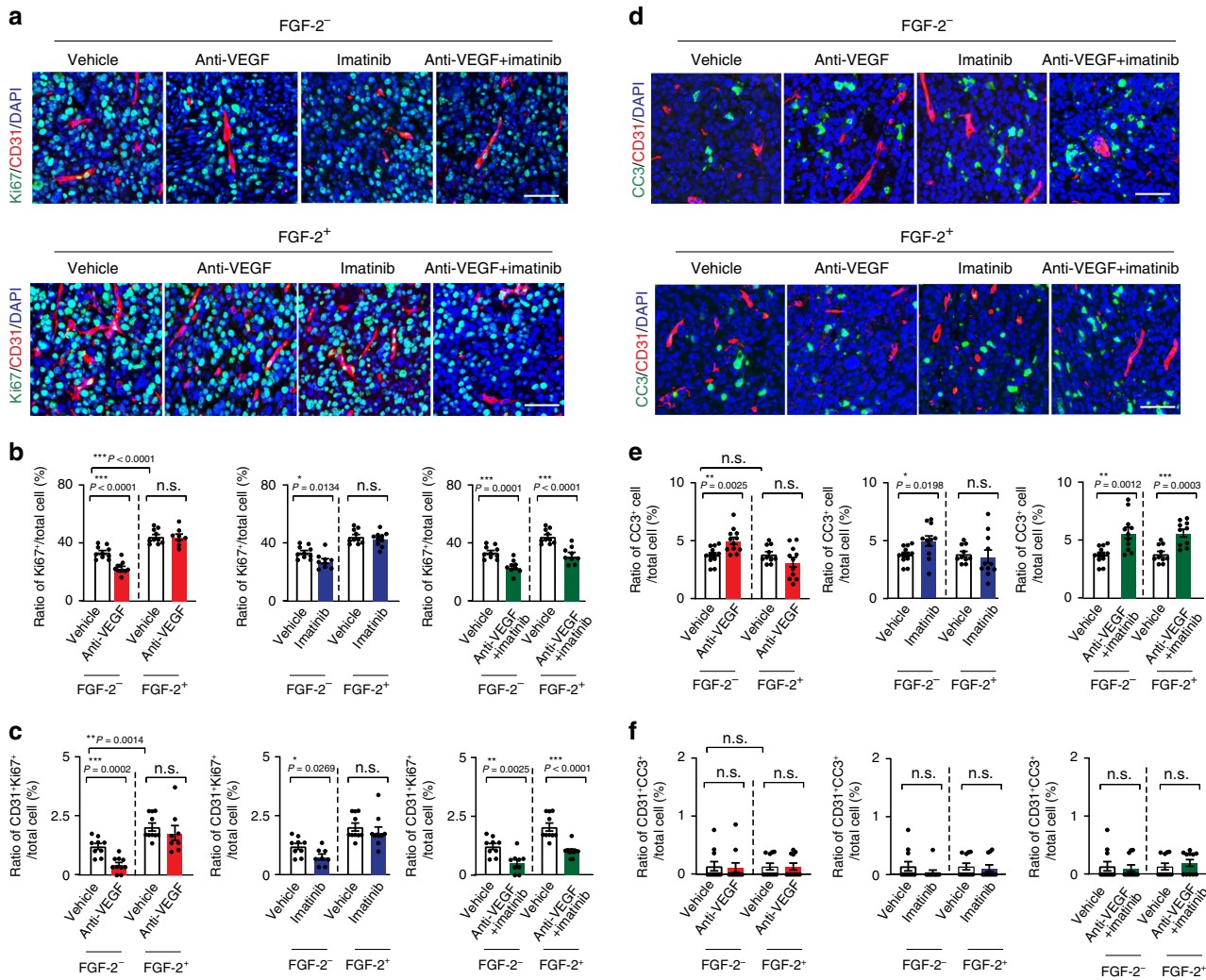

**Fig. 5 Tumor cell proliferation and apoptosis of FGF-2 fibrosarcomas in response to various drug treatment. a** Ki67+ proliferative cell signals (green) co-stained with CD31+ microvessels (red) and DAPI (blue) of various monotherapy and combination therapy-treated T241-vector and T241-FGF-2 fibrosarcomas. Bar = 50 μm. **b** Quantification of Ki67+ signals in vehicle-, anti-VEGF-, imatinib- and combination therapy-treated T241-vector and T241-FGF-2 fibrosarcomas ($n$(Vector) = 10/11/9/9; n(FGF-2) = 10/8/10/9; $P$(Vector vs FGF-2) < 0.0001; $P$(Vehicle-treated vector vs anti-VEGF-treated vector) < 0.0001; $P$(Vehicle-treated vector vs imatinib-treated vector) = 0.0134; $P$(Vehicle-treated vector vs the combination therapy-treated vector) < 0.0001; $P$(Vehicle-treated FGF-2 vs the combination therapy-treated FGF-2) < 0.0001. **c** Quantification of Ki67+ and CD31+ double-positive signals in vehicle-, anti-VEGF-, imatinib- and combination therapy-treated T241-vector and T241-FGF-2 fibrosarcomas ($n$(Vector) = 9/10/8/9; n(FGF-2) = 10/8/10/10; $P$(Vector vs FGF-2) = 0.0014; $P$(Vehicle-treated vector vs anti-VEGF-treated vector) = 0.0002; $P$(Vehicle-treated vector vs imatinib-treated vector) = 0.0269; $P$(Vehicle-treated vector vs combination therapy-treated vector) = 0.0025; $P$(Vehicle-treated FGF-2 vs combination therapy-treated FGF-2) < 0.0001. **d** Micrographs of caspase-3+ apoptotic cells (green) co-stained with CD31+ microvessels (red) and DAPI (blue) in various monotherapy- and combination therapy-treated T241-vector and T241-FGF-2 fibrosarcomas. Bar = 50 μm. **e** Quantification of caspase-3+ signals in vehicle-, anti-VEGF-, imatinib- and combination therapy-treated T241-vector and T241-FGF-2 fibrosarcomas ($n$(Vector) = = 12/12/11/11; n(Vector) = 10/11/11/10; $P$(Vehicle-treated vector vs anti-VEGF-treated vector) = 0.0025; $P$(Vehicle-treated vector vs imatinib-treated vector) = 0.0198; $P$(Vehicle-treated vector vs combination therapy-treated vector) = 0.0012; $P$(Vehicle-treated FGF-2 vs combination therapy-treated FGF-2) = 0.0003). **f** Quantification of caspase-3+ -CD31+ double-positive signals in vehicle-, anti-VEGF-, imatinib- and combination therapy-treated T241-vector and T241-FGF-2 fibrosarcomas ($n$(Vector) = 11 each; n(Vector) = 11 each). FGF-2− = vector cancers; FGF-2+ = FGF-2 cancers; n.s. Not significant; *$P$ < 0.05; **$P$ < 0.01; ***$P$ < 0.001; two-tailed $t$-test. Data presented as mean ± s.e.m. Experiments were independently repeated twice. Source data are provided as a Source Data file.

chemotherapy might produce enhanced antitumor activity. These findings warrant further validation in clinical settings.

**Systemic effects of combination therapy.** Systemic delivery of anti-VEGF drugs has previously been shown to regress microvasculatures in multiple tissues and organs. We next studied the systemic impact of anti-VEGF + imatinib combination therapy on microvessels in healthy tissues. We should emphasize that a relative low dose of VEGF blockade (2.5 mg kg⁻¹) has been used in the

combination therapy. The reason for deliberately choosing the low dose was to investigate the synergistic anticancer effects by these two drugs. Among the examined tissues, including thyroid, kidney and skeletal muscle tissues, a significant reduction of microvessel density was observed in thyroid (Supplementary Fig. 9). However, microvascular density in kidney and skeletal muscle tissues were unchanged after the combination therapy (Supplementary Fig. 9). Consistent with reduction of thyroid microvasculatures, blood perfusion of 2000-kDa fluorescein-lysinated dextran was also

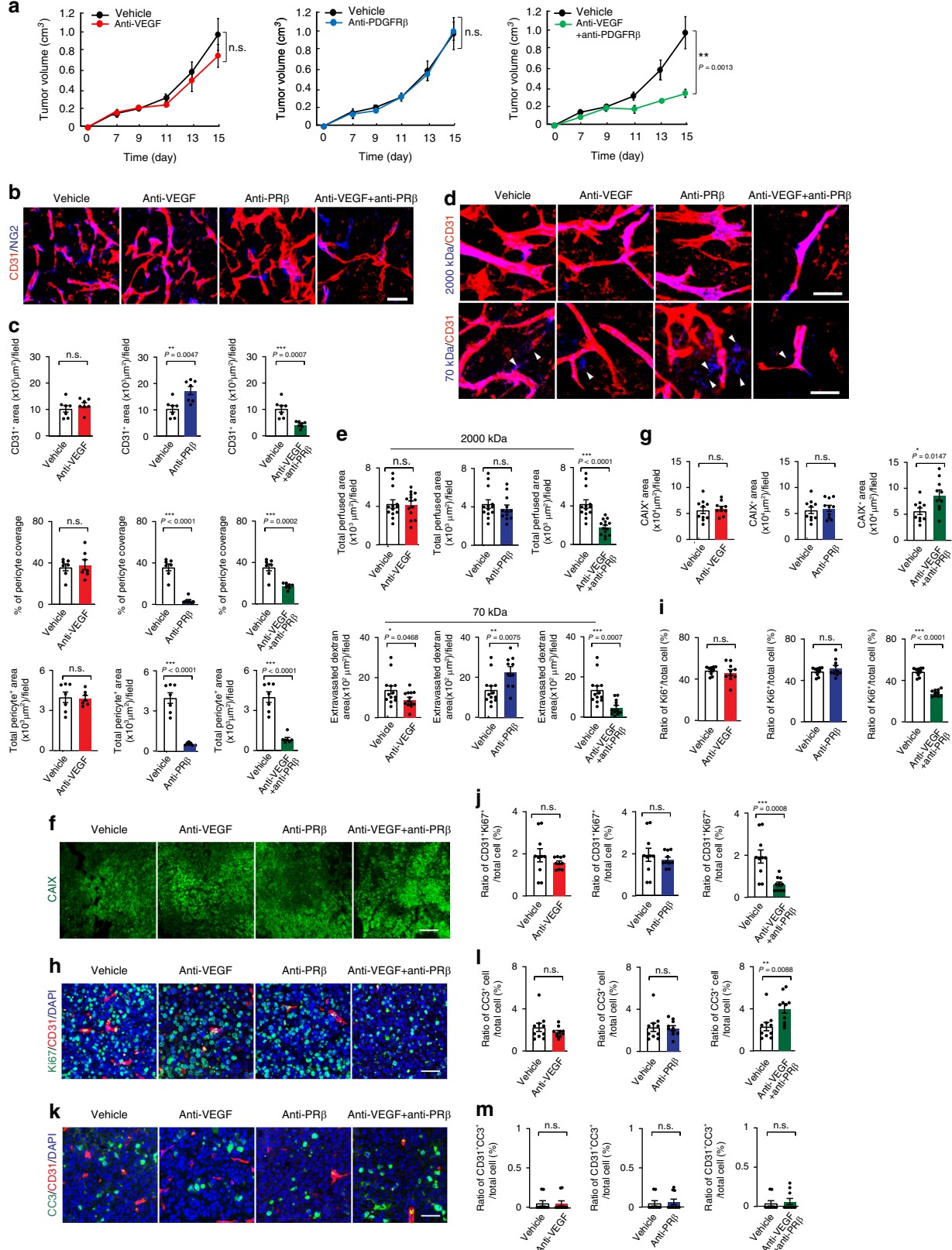

significantly decreased in thyroid, but not decreased in kidney and skeletal muscles (Supplementary Fig. 9). Pericyte coverage and leakiness of 70-kDa fluorescein-lysinated dextran were not altered. These findings are consistent with previous reports that endocrine vasculatures are highly sensitive to systemic anti-VEGF therapy.

The anti-VEGF + imatinib combination therapy-treated mice showed no differences in body weight and proteinuria (Supplementary Fig. 9). Slightly increases of systolic and diastolic blood pressures were observed in the combination therapy-treated animals (Supplementary Fig. 9).

**Fig. 6 Impact of anti-PDGFRβ on tumor growth, angiogenesis, and tumor microenvironment. a** Tumor growth of T241-FGF-2 fibrosarcomas in response to anti-VEGF, anti-PDGFRβ, and anti-VEGF plus anti-PDGFRβ treatments ($n = 7/6/6/7$; $P$(Vehicle vs combination therapy) = 0.0013). Time indicates after the start of treatment. **b** Microvascular density and perivascular coverage of anti-VEGF-, anti-PDGFRβ-, and anti-VEGF plus anti-PDGFRβ-treated T241-FGF-2 fibrosarcomas. Red indicates CD31$^+$ microvessels and blue indicates NG2$^+$ pericytes. Bar = 50 μm. **c** Quantification of CD31$^+$ microvessel density ($n = 7$ each; $P$(Vehicle vs anti-PDGFRβ) = 0.0047; $P$(Vehicle vs combination) = 0.0007), NG2$^+$ pericyte coverage ($n = 7$ each; $P$(Vehicle vs anti-PDGFRβ) < 0.0001; $P$(Vehicle vs combination) = 0.0002), and NG2$^+$ pericyte area ($n = 7$ each; $P$(Vehicle vs anti-PDGFRβ) < 0.0001; $P$(Vehicle vs combination therapy) < 0.0001) of various therapy-treated fibrosarcomas. **d** Micrographs of vascular perfusion of 2000 kDa dextran (blue) and vascular permeability of 70 kDa dextran (blue) co-stained with CD31$^+$ microvessels (red). Bar = 50 μm. **e** Quantification of perfused vasculature ($n = 13/14/12/14$; $P$(Vehicle vs combination) < 0.0001) and vascular leakiness ($n = 13/13/10/12$; $P$(Vehicle vs anti-VEGF) = 0.0468; $P$(Vehicle vs anti-PDGFRβ) = 0.0075; $P$(Vehicle vs combination) = 0.0007) of various therapy-treated fibrosarcomas. **f** CAXI$^+$ signals of tumor hypoxia. Bar = 100 μm. **g** Quantification of CAXI$^+$ hypoxic signals of various therapy-treated fibrosarcomas ($n = 10$ each; $P$(Vehicle vs combination) = 0.0147). **h** Ki67$^+$ proliferative cell signals of various therapy-treated fibrosarcomas. Bar = 50 μm. **i** Quantification of Ki67$^+$ signals in various therapy-treated fibrosarcomas ($n = 10$ each; $P$(Vehicle vs combination) < 0.0001). **j** Quantification of Ki67$^+$ and CD31$^+$ double-positive signals in various therapy-treated fibrosarcomas ($n = 10$ each; $P$(Vehicle vs combination therapy) = 0.0008). **k** Micrographs of caspase-3$^+$ apoptotic cells in various therapy-treated fibrosarcomas. Bar = 50 μm. **l** Quantification of caspase-3$^+$ signals in various therapy-treated fibrosarcomas ($n = 10$ each; $P$(Vehicle vs combination therapy) = 0.0088). **m** Quantification of caspase-3$^+$-CD31$^+$ double-positive signals in various therapy-treated fibrosarcomas ($n = 10$ each). anti-PRβ = anti-PDGFRβ; n.s. Not significant; two-tailed student $t$-test. Data presented as mean ± s.e.m. Experiments were repeated twice. Source data are provided as a Source Data file.

## Discussion

There are a few key unsolved issues that impede clinical benefits of AADs in human cancer patients, including understanding of drug resistance, selection of reliable biomarkers, and timeline of treatment, and optimization of combination therapy[5]. At this time of writing, these issues remain elusive and have no universal solutions for improving therapeutic benefits of these drugs. In our opinion, understanding the fundamental mechanisms that underlie tumor angiogenesis, vascular remodeling, and the drug-induced alterations of TME is crucial for therapeutic improvement.

Our present work unravels a mechanism of improving therapeutic benefits by combining two drugs that do not target the predominantly expressed vascular factor in tumors. FGF-2 has long been known as a potent angiogenic factor that stimulates tumor neovascularization[33–35]. By TCGA analysis, we show that FGF-2 expression levels reversely and significantly correlated with survival advantages in breast, ovarian, and bladder cancers (Fig. 7a–d). In this study, we show that FGF-2 displays a robust effect on remodeling tumor microvasculatures by increasing perivascular contents and coverage. Both stimulation of angiogenesis and recruitment of pericytes were overwhelmed in FGF-2$^+$ tumors. Our recent findings show that the FGF-2-FGFR2 signaling in pericytes mediates cell proliferation and amplifies the pool pericytes in tumors[24]. Intriguingly, the FGF-2-FGFR1 signaling in endothelial cells induces expression of PDGF-B and PDGF-D to ensure activation of perivascular PDGFRβ, which is the common receptor for PDGF-B and PDGF-D and crucial for pericyte recruitment[24]. Thus, FGF-2 plays dual roles in angiogenesis and vascular remodeling through endothelial FGFR1 and pericytic FGFR2. For tumor angiogenesis, FGF-2 seems to collaborate with the VEGF-VEGFR2 signaling to ensure both processes of endothelial cell proliferation and sprouting simultaneously occur. While the VEGF-triggered signaling plays a guidance role of vascular patterning[16], FGF-2 induces endothelial proliferation for establishing the sufficient number of microvessels[25]. For pericyte coverage, FGF-2 amplifies the PDGF-B-PDGFRβ signaling at both ligand and receptor levels to ensure appropriate remodeling and maturation of tumor vessels (Fig. 7e)[19,24]. Therefore, intimate and orchestrated interplay between FGF-2, VEGFs, and PDGFs necessitate the establishment of functional vasculatures in tumors.

Consequently, blocking each of these signaling pathways alone would not sufficiently inhibit tumor angiogenesis. Expectedly, FGF-2 tumors are completely resistant to the anti-VEGF treatment, supporting the compensatory concept of AAD resistance. Surprisingly, inhibition of the PDGFR signaling alone in FGF-2$^+$

tumors further augments neovascularization. A possible mechanistic explanation of the elevated angiogenic response would attribute ablation of vascular pericytes by PDGFRβ inhibition, allowing further exposure of tumor vessels to FGF-2 and VEGF, and consequently leading to excessive sprouting of microvessels (Fig. 7e). However, the anti-PDGFRβ-augmented tumor angiogenesis did not instigate accelerated tumor growth, suggesting the non-productive features of these vessels. Alternatively, inhibition of PDGFRβ might induce the expression of angiopoietin-like 4 (Angptl4), which augments angiogenesis[36]. Also, we cannot exclude the possibility that anti-VEGF and imatinib might influence the promoter activity of the vector that drives FGF-2 expression. Another interesting finding from this study is that pan-blocking FGFRs did not result in an expectedly potent antitumor effect on FGF-2$^+$ tumors (Supplementary Fig. 10). These data indicate that other angiogenic factors such as VEGF could circumvent the FGF blockade. If none of these monotherapeutics produce robust antitumor effects, why would dual blocking VEGF and PDGF produce a synergistic and robust antitumor effect on FGF-2 tumors? A rationalized explanation would be FGF-2-induced angiogenesis and vascular remodeling are dependent on VEGF and PDGF and simultaneous inhibition of these factors not only blocks their own signaling pathways, but also blocks the FGF-2-mediated vascular effects. We did not see improved blood perfusion in any of these drug-treated tumors. For therapeutic intervention, the VEGF-VEGFR signaling should always be taken into consideration because of its non-replaceable role of angiogenesis in tumors.

Defining reliable biomarkers to predict AAD therapeutic benefits has become one of the most challenging issues for anti-angiogenic cancer therapy[5,37,38]. Bevacizumab as a monospecific drug only targets VEGF and ironically VEGF expression levels in tumors cannot be used as a predictive marker[39]. Why would the monospecific VEGF target not be used as a predictive marker? At this time of writing, this issue remains an enigma. Our present work proposes a concept of off-drug targets as a potential predictive marker for therapeutic benefits. This paradigm has changed our way of thinking in defining predictive biomarkers for antiangiogenic therapy. Our work can be further extended to circumvent drug resistance by off-targeted combination therapy. Combination of AAD drugs with different principles may block common pathways of tumor angiogenesis and vascular functions. We use FGF-2 as an example to demonstrate the principle of combination therapy to circumvent drug resistance. This principle may also apply for improving therapeutic efficacy of drugs targeting other angiogenic factors.

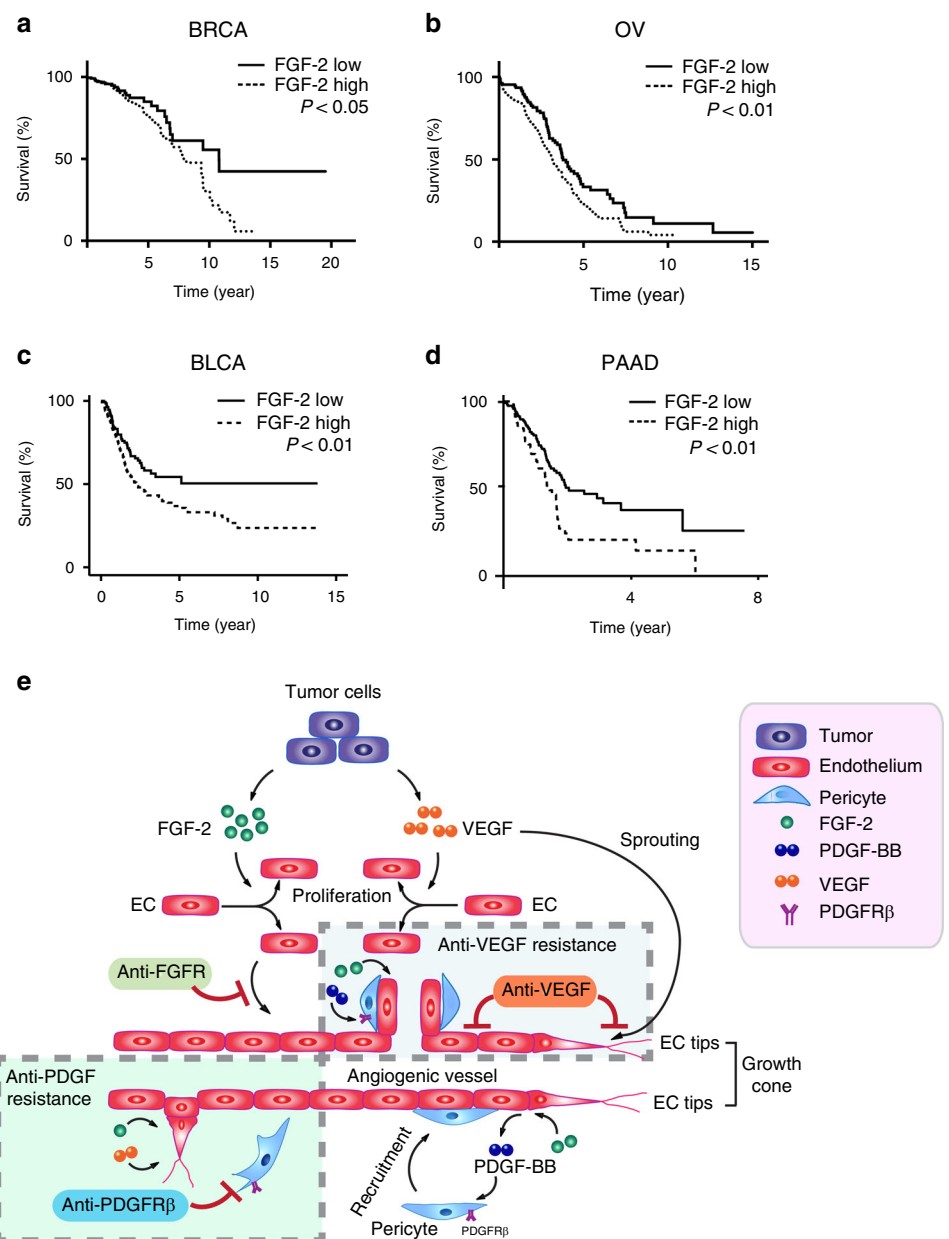

**Fig. 7 Survival correlation and schematic diagram of underlying mechanisms of synergistic anti-FGF-2$^+$ tumor activity by combination therapy.**
**a** Kaplan–Meier survival of FGF-2-high vs. FGF-2-low breast cancer (BRCA, $n = 246$ vs. 267; $P = 0.0458$); **b** ovarian cancer (OV, $n = 131$ vs. 129; $P = 0.0097$); **c** bladder carcinoma (BLCA, $n = 261$ vs. 141; $P = 0.007$); **d** and pancreatic adenocarcinoma (PAAD, $n = 57$ vs. 120; $P = 0.003$). The log-rank test was used for statistical analysis at the endpoint. Source data are provided as a Source Data file. **e** Tumors often produce FGF-2 and VEGF to stimulate tumor angiogenesis. While VEGF stimulates endothelial cell proliferation, migration, and endothelial cell tip formation, FGF-2 primarily induces endothelial cell proliferation. In FGF-2 positive tumors, blocking FGF-2-triggered signaling such as inhibition of FGFR inhibits endothelial cell proliferation and angiogenesis. However, the impact of anti-FGF agents on tumor angiogenesis may be modest because tumors employ VEGF to stimulate neovascularization. Thus, VEGF plays a compensatory role in circumventing the antiangiogenic effect by FGF-2. Similarly, blocking VEGF alone in FGF-2 positive tumors may also produce a limited antitumor effect because of the compensatory effect of FGF-2. In addition, FGF-2 is a potent perivascular factor to stimulate pericyte proliferation and vascular coverage through an intimate collaboration with the PDGF-B-PDGFRβ signaling pathway. Blocking PDGFRβ alone would lead to ablation of perivascular cells from tumor vessels, permitting exposure of endothelial cells to vascular stimuli such as FGF-2 and VEGF. In supporting this view, we show that anti-PDGFRβ increases rather than reduces vascular density in FGF-2 positive tumors. Simultaneous blocking VEGF and PDGFRβ signaling pathways inhibits vascular sprouting and vascular stability, leading to vascular regression in FGF-2 positive tumors.

Our findings have profound clinical implications of defining predictive biomarkers for improving antiangiogenic therapy and for circumventing drug resistance. Combination of AADs with different mechanistic principles may significantly improve therapeutic outcomes. This conceptual paradigm warrants clinical approvals in human cancer patients by designing rigorous clinical trials.

## Methods

**Cell and cell culture**. Murine breast cancer E0771 cell line purchased from CH3 BioSystems; murine fibrosarcoma T241 cell line purchased from ATCC. E0771 cells were cultured in RPMI-1640 medium (Sigma) supplemented with 10% fetal bovine serum (FBS) (Gibco), 100 U mL$^{-1}$ penicillin, and 100 µg mL$^{-1}$ streptomycin (SV30010; HyClone). T241 cells grown in Dulbecco's modified Eagle's medium (DMEM) (Sigma) supplemented with 10% FBS, 100 U mL$^{-1}$ penicillin, and 100 µg mL$^{-1}$ streptomycin (SV30010; HyClone). All cells were

maintained at 37 °C in a 5% $CO_2$-humidified incubator. Mycoplasma was analyzed to ensure germ-free. E0771 and T241 cell lines were used for establishing stable enhanced green fluorescent protein (EGFP)- alone and EGFP–hFGF-2-expressing cell lines. An *FGF-2* cDNA coding for the full-length human FGF-2 fused to a human an *Il2* secretory signal sequence was cloned into a construct containing pMXs-internal ribosome entry site-GFP (pMXs-IG) (a gift from Dr. Toshio Kitamura, The Institute of Medical Science, Tokyo, Japan), generating the pMXs-IG-FGF-2-CS23 construct. The specific primer pairs used were forward: 5′-TCAGCTCTTAGCAGACATTG-3′ and reverse: 5′-GGGAATTCGCCACCATG-TACAGGATGCAACTCCTCTCTTGCATTGCACTAAGTCTTGCAC-3′. E0771 and T241 cells were transfected with the pMXs-IG-FGF-2-CS23 construct by Lipofectamin2000 according to manufacturer's instruction. Transfected GFP-positive Vector or FGF-2 cells were selected using flow cytometry (MoFlo XTD; Beckman Coulter). Murine primary endothelial cells and pericytes were isolated by FACS using antibodies against CD31(553370; BD Pharmingen) and NG2 (AB5320; Merck). Cells were maintained in DMEM supplemented with 10% FBS, 100 U mL$^{-1}$ penicillin, and 100 µg mL$^{-1}$ streptomycin.

**Mouse tumor models and drug treatment.** Animal studies were approved by the North Stockholm Animal Ethical Committee in Sweden. Wild-type C57BL/6 mice were obtained from the animal facility at the Department of Microbiology, Tumor and Cell Biology, Karolinska Institute, Sweden. Animal studies were approved by the animal Experimental Ethical Committee of Fudan University in China and wild-type C57BL/6 mice were obtained from the Model Animal Research Center of Nanjing University. Mice were maintained in the animal facility with a constant temperature (20 ± 2 °C), constant humidity (50 ± 10%), and a 12 h: 12 h light/dark cycle. Age- and sex-matched mice were randomly divided to each group. Approximately $1 \times 10^6$ T241-vector or -FGF-2 tumor cells were subcutaneously injected into the back along the mid dorsal line of each C57BL/6 mouse. Approximately $0.5 \times 10^6$ E0771-vector or -FGF-2 tumor cells were injected into the mammary fat pad in each C57BL/6 female mouse. Tumor volume was measured and calculated according to the standard formula (length × width$^2$ × 0.52). Treatment was initiated at day 5 after tumor injection and terminated when the average tumor size in the control group reached the size of 1.0–1.2 cm$^3$. In another experimental setting, treatment of the established tumors (0.2–0.3 cm$^3$ size) was terminated when the average tumor volume of the control group reached 1.4 cm$^3$. A rabbit anti-mouse VEGF neutralizing antibody (2.5 mg kg$^{-1}$; BD0801, Nanjing, China, kindly provided by the Simcere Pharmaceutical Company) and a rat anti-mouse PDGFRβ neutralizing antibody (40 mg kg$^{-1}$; 2C5, ImClone Pharmaceuticals, kindly provided by Dr. Zhenping Zhu) were intraperitoneally injected twice per week in either monotherapy or combination settings. Imatinib (50 mg kg$^{-1}$, LC laboratories) was intraperitoneally injected daily. Vehicle-treated mice served as controls. For chemotherapy, 5-FU (Roche) at a low dose of 10 mg kg$^{-1}$ and a high dose of 60 mg kg$^{-1}$ was intraperitoneally injected once a week alone or in combination with an anti-mouse VEGF neutralizing antibody plus imatinib. A pan-FGFR inhibitor, BGJ398 (Novartis), was prepared in an acetic acid buffer (pH 4.6) for 10 min, followed by mixing with PEG300 (50% of final volume) using vortex. BGJ398 was orally administrated daily to each mouse at a dose of 30 mg kg$^{-1}$. Non tumor-bearing mice were treated with a vehicle and anti-VEGF plus imatinib combination for 2 weeks. Animal numbers in each group were indicated in each figure legend. All animal experiments except blood perfusion and vascular permeability were terminated using a lethal dose of $CO_2$.

**Whole-mount staining.** Tumor tissue samples or healthy tissues from non-tumor-bearing mice were collected and fixed overnight with 4% PFA. Tissue samples were cut into thin pieces and digested for 5 min with 20 mM proteinase K in a 10 mM Tris-buffer (pH 7.5), followed by incubation with 100% methanol for 30 min. Samples were incubated overnight at 4 °C in 0.3% Triton X-100 PBS containing 3% skim milk. After washing several times with PBS, tissue samples were incubated with a combination of a rat anti-mouse CD31 (1:200; 553370; BD Pharmingen) antibody and a rabbit polyclonal anti-NG2 (1:200; AB5320; Merck) antibody, followed by incubation with secondary antibodies, consisting of an Alexa Fluor 555-labeled goat anti-rat (1:200; A21434; Thermo Fisher SCIENTIFIC) and an Alexa Fluor 647-labeled goat anti-rabbit (1:200; A-21244; Thermo Fisher SCIENTIFIC). After washing with PBS, tissues were mounted using a vectashield-mounting medium (H1000; Vector Laboratories) and images were taken by confocal microscopy (Nikon C1 Confocal microscope, Nikon Corporation, Japan). Three-dimensional images of tumor vessels were analyzed. Positive signals of CD31 or NG2 area were calculated using an Adobe Photoshop software (CS6; Adobe) program. Pericyte coverage was quantified as a percentage of vessel areas covered by pericytes with a calculation of overlap area of CD31 and NG2 positive signals.

**Blood perfusion and vascular permeability.** Prior to termination of experiments, 0.5 mg of 2000-kDa-lysinated-rhodamine dextran (LRD) (D7139; Thermo Fisher SCIENTIFIC) or 2000-kDa-lysinated fluorescein dextran (LFD) (D7137; Thermo Fisher SCIENTIFIC), or 0.6 mg of 70-kDa-LRD (D1818; Thermo Fisher SCIENTIFIC) or 70-kDa-LFD (D1822; Thermo Fisher SCIENTIFIC) in 100 µl dH$_2$O was intravenously injected into the tail vein of each mouse. At 5-min or 15-min post injection, mice were euthanized by cervical dislocation or inhalation of high dose of

isoflurane. Tumors and healthy tissues of non-tumor-bearing mice were removed and were fixed overnight with 4% PFA solution, followed by whole-mount immunostaining as described above. A rat anti-mouse CD31 (1:200) or a goat anti-mouse CD31 (1:200; AF3628; R&D SYSTEMS), a Cy5-labeled goat anti-rat secondary antibody (1:200; AP183S; Invitrogen), and an Alexa Fluor 555-labeled donkey anti-goat secondary antibody (1:200; A-21432; Invitrogen) were used for visualization of the vasculatures. LRD or LFD, and vascular positive signals were detected by confocal Microscopy (Nikon C1 Confocal microscope; Nikon Corporation, Japan) and three-dimensional images were collected. Vascular perfusion was quantified as a vessel area from the 2000-kDa-LRD/LFD positive signals per field and extravasation was quantified from the 70-kDa-LRD/LFD area extravasated from vessels per field.

**Immunohistochemistry.** Paraffin-embedded tumor issues were cut into 5-µm thick section. After baking, tissue slides were de-paraffinized in Tissue-Clear (1466; Sakura) and rehydrated with sequential steps in 99–95–70% ethanol followed by boiling for 20 min in an unmasking solution (H3300, VECTOR), and subsequently blocked with 4% serum. Cryopreserved tissues were cut into 5–10 µm thick section. After fixing with 4% PFA, sections were washed three times in PBS, followed by incubation with 4% serum. Tissue slides were stained with a goat anti-mouse CD31 (1:400; AF3628; R&D systems) antibody, an anti αSMA (1:200; M0851; clone 1A4; DAKO) antibody, a rabbit anti-mouse Cleaved Caspase 3 (1:200; 9661; Cell Signaling) antibody, and a rabbit anti-mouse Ki67 (1:200; PA5-19462; Thermo Fisher SCIENTIFIC) antibody, a rabbit anti-mouse FSP1 (1:300; 07-2274; Merck) antibody, a rabbit anti-mouse Iba1 (1:200; DAKO; 019-19741) antibody, a rat anti-mouse CD3 (1:50, Invitrogen;17-0032-80) antibody, a rat anti-mouse CD4 (1:50; BD Pharmingen; 550280) antibody, and a rat anti-mouse CD8 (1:50; BD Pharmingen; 550281) antibody, followed by staining with species-matched secondary antibodies as follows: an Alexa Fluor 555-labeled donkey anti-goat (1:400; A21432; Thermo Fisher SCIENTIFIC), an Alexa Fluor 488-labeled donkey anti-mouse (1:400; A21202; Thermo Fisher SCIENTIFIC), an Alexa Fluor 488-labeled donkey anti-rabbit (1:400; A21206, Thermo Fisher SCIENTIFIC), or an Alexa Fluor 555-labeled goat anti-rat (1:400; A21434; Thermo Fisher SCIENTIFIC) antibody. For detection of tumor hypoxia, a rabbit anti-CAIX antibody (1:400; NB100-417; NOVUS) followed by an Alexa Fluor 488-labeled donkey anti-rabbit antibody (1:400; A21206, Thermo Fisher SCIENTIFIC), or a FITC-conjugated mouse anti-pimonidazole antibody was used. In the latter case, 1.5 mg pimonidazole (Hypoxyprobe) in 100 µl PBS was intraperitoneally injected into each mouse, followed by sacrifice 30 min later. A FITC-conjugated mouse anti-pimonidazole monoclonal antibody (clone 4.3.11.3, Hypoxyprobe) was used in paraffin tissues. Positive signals were detected using a fluorescence microscope equipped with a camera (Nikon, DS-QilMC). Images were analyzed using an ImageJ and an Adobe Photoshop software (CS6; Adobe) program.

**Tumor necrosis assay.** The paraffin-embedded tumor tissues were cut into 5-µm-thick sections deparaffinized in Tissue-Clear (Cat. No. 1466, Sakura), and rehydrated with 99–95–70% ethanol using a stepwise procedure. Tissue slides were stained with hematoxylin (6765009; Thermo Fisher SCIENTIFIC), followed by eosin (HT110116; Sigma). After dehydrated with 95–99% ethanol, slides were mounted with PERTEX (Cat. No. 00801, HistoLab). Stained tumor tissues were photographed using a light microscope (Nikon Eclipse TS100) equipped with a camera (DS-Fi1; Nikon) and software (NIS-Elements F3.0). Multiple pictures collected from each tissue sample were integrated and total massive necrosis in tumors analyzed using a Photoshop software (CS6; Adobe).

**ELISA.** Enzyme-linked immunosorbent assay was used for measuring human FGF-2, mouse VEGF, and mouse PDGFB protein concentrations. To detect protein levels in tissues and cells, tumor tissues and cells in culture were homogenized in a lysis buffer (C3228, Sigma), containing proteinase inhibitor cocktails (1:100, P8340, Sigma), followed by 15 min centrifugation. The supernatants were transferred to a new collection tube and either immediately used or stored at −20 °C or −80 °C until further analysis. Conditioned media were collected from confluent E0771 and T241 cells at 24 and 48 h. Cell debris was removed by centrifugation and supernatants were stored at −20 °C or −80 °C until analysis. The human FGF-2 (DFB50, R&D systems), mouse VEGF (MMV00), and mouse PDGF-BB ELISA assays (MBB00, R&D Systems) were performed according to the manufacturer's protocol with the standard curve. Absorbance values were detected at 450 nm using a microplate reader and values were calculated using the formula obtained from the trendline.

**Correlation of FGF-2 expression and survival in patients.** Survival data of 513 breast invasive carcinoma (BRCA), 260 ovarian cancer (OV), 402 urothelial bladder carcinoma (BLCA), and 177 pancreatic adenocarcinoma (PAAD) patients from The Cancer Genome Atlas (TCGA) were analyzed for FGF-2-high (BRCA; top 30%, OV; above median, BLCA and PAAD; above top 35%) and FGF-2-low (BRCA; lowest 10%, OV; below median, BLCA and PAAD; below 65%) groups. The statistical differences were analyzed using the Kaplan–Meier survival method followed by a log-rank test. The Cancer Genome Atlas data sets for the survival study are downloaded from the link; http://gdac.broadinstitute.org/runs/

**Cell proliferation assay.** Tumor cells ($2-3 \times 10^3$), endothelial cells ($2 \times 10^3$), and pericytes ($1 \times 10^3$) were seeded onto each well of a 96-well plate. Proliferation of tumor cells treated with indicated doses of VEGF blockade, PDGFRβ blockade, and a control vehicle was analyzed at 24, 48, 72, and 96 h using an MTT (5 mg mL$^{-1}$; M5655; Sigma-Aldrich) assay. Endothelial cells and pericytes were incubated with conditioned media collected from tumor cell culture diluted in fresh DMEM supplemented with 2% FBS. Cell proliferation was measured at 72 h. Recombinant human FGF-2- (100-18 C; PEPROTECH) treated endothelial cells (5 ng mL$^{-1}$) and pericytes (50 ng mL$^{-1}$) were used as positive controls. Absorbance values at 490 nm were obtained using a spectrophotometer (FLUOstar Omega; BMG LABTECH).

**Urine protein electrophoresis.** Spot urine samples from spontaneous bladder voiding were collected before the termination of experiments and were stored at $-20\,°C$ or $-80\,°C$ until further analysis. Urine samples were centrifuged at $11,200 \times g$ for 10 min. A mixture of a 5 µl urine sample and a triton-based lysis buffer (C3228, Sigma) was heated at 100 °C for 5 min. An equal amount of each sample was applied to a SDS-PAGE gel (NP0321/NP0323, Life Technologies), followed by the incubation for 30 min with 50% MeOH, 10% HoAC, and 40% $H_2O$. Gels were subsequently stained with the Coomassie Brilliant blue dye (24620; Thermo Fischer SCIENTIFIC) for 1 h followed by 2-h incubation with 5% MeOH, 7.5% HoAC, and 87.5% $H_2O$. An Odyssey CLx system (LI-COR) equipped with an Image Studio software was used for photography and data analysis. The original blot image is presented in Supplementary Fig. 11.

**Blood pressure measurement.** Systolic and diastolic blood pressures were measured from mice tail vein using a noninvasive CODA tail-cuff blood pressure system with the volume pressure R\recording (VPR) technique (CODA HT2, Kent Scientific). Each mouse was located in a small chamber on a 37 °C plate and accustomed for a while before a set of measurement. After putting the tail-cuff, blood pressure measurement was performed for 25 cycles and an average pressure excluding the error measurements (i.e. due to mice movement) was presented as a value for each mouse. ($n = 4-5$ mice per group).

**FACS analysis.** Fresh tumor tissues were cut into small pieces and were incubated in a combination of 0.15% type I collagenase (Sigma-Aldrich; catalog no. C0130-500MG) and 0.15% type II collagenase (Sigma- Aldrich; catalog no. C6885-1G) at 37 °C for 1 h. Digested tissues were filtered with a 100-µm cell strainer, followed by a 70 µm cell strainer. Single cells from each sample after centrifugation were stained on ice for 15 min with an Alexa Fluor 647–conjugated anti-mouse F4/80 antibody (123122, BioLegend), an APC–conjugated anti-mouse CD3 antibody (17-0032-80, BioLegend), an APC–conjugated anti-mouse CD4 antibody (100411, BioLegend), and a PerCP–conjugated anti-mouse CD8 antibody (100731, BioLegend). The stained samples filtered with a 40-µm cell strainer were scanned on a FACScans (Becton Dickinson) and data were analyzed with a CellQuestPro software (BD Biosciences). The gating strategy of flow cytometry is presented in Supplementary Fig. 11.

**Statistical analysis.** Collected data were analyzed using a Microsoft Excel and a GraphPad software. Data presented means ± SEM. Statistical analysis was performed using the standard two-tailed Student $t$-test, and $*P < 0.05$, $**P < 0.01$, and $***P < 0.001$ were considered statistically significant.

**Reporting summary.** Further information on research design is available in the Nature Research Reporting Summary linked to this article.

## Data availability
Data supporting the findings of this study are available within the article and its Supplementary Information files. The Cancer Genome Atlas data sets for the survival study are downloaded from the link; http://gdac.broadinstitute.org/runs//. The values used for the survival study are provided as a Source Data file (Fig. 7a–d). Source data are provided with this paper.

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

## Acknowledgements

Y.C.'s laboratory is supported through research grants from the European Research Council (ERC) advanced grant ANGIOFAT (Project no 250021), the Swedish Research Council, the Swedish Cancer Foundation, the Swedish Children's Cancer Foundation, the Strategic Research Areas (SFO)–Stem Cell and Regenerative Medicine Foundation, the Karolinska Institute Foundation, the Karolinska Institute distinguished professor award, the Torsten Soderbergs Foundation, the Maud and Birger Gustavsson Foundation, the NOVO Nordisk Foundation-Advance grant, and the Knut and Alice Wallenberg's Foundation. J.X. is supported by the National Natural Science Foundation of China (project 81801163). We thank Novartis for providing BGJ398. Open access funding provided by Karolinska Institute.

## Author contributions

Y.C. designed the project. Y.C. and K.H. planned the experiments and analyzed the data. K.H., Y.Y., T.S., Q.D., J.X., J.W., Y.Z, M.N., M.S., and X.S. participated in experimentation. X.H. and H.M. performed the survival analysis. Y.X., T.C., X.L., X.Li., Q.L., Y.L., A.H., and Y.Ch. participated in the discussion. Y.C. wrote the manuscript.

## Competing interests

The authors declare no competing interests.
