## [Peer Review File · Nature Communications]

Reviewers' Comments:

Reviewer #1:

Remarks to the Author:

This paper is of broad interest to the readership of Nature Communications for the following reasons

- 1) It identifies FGF2 as a potential biomarker of Avastin resistant tumors
- 2) It identifies FGF2 as a mechanism of resistance
- 3) It demonstrates that the combination of VEGF blockade and imatinib are effective against FGF positive tumors, even when the monotherapy of each one is ineffective. This should lead to reconsideration that if a drug is ineffective in a given clinical setting, it may contribute to the effectiveness of a combination therapy. This may lead to a general re-evaluation of how we determine drug efficacy in clinical trials
- 4) Imatinib inhibits FGF signaling despite not inhibiting FGF receptor signaling.

Specific comments: Imatinib monotherapy appears to induce a hypervascular and hyperpermeable phenotype compared with other treatments. Please comment on this. One potential mechanism might be the induction of ANgpt4, and Angpt4 blockade with propranolol might ameliorate this phenomenon- see Propranolol exhibits activity against hemangiomas independent of beta blockade.

NPJ Precis Oncol. 2019 Nov 1;3:27.

- 2) Please reference the following articles on PDGF signaling

(Cooperative benefit for the combination of rapamycin and imatinib in tuberous sclerosis complex neoplasia.

Vasc Cell. 2012 Jul 5;4(1):11. doi)

(Malignant transformation of human cells by constitutive expression of platelet-derived growth factor-BB.

J Biol Chem. 2005 Apr 8;280(14):13936-43.)

Reviewer #2:

Remarks to the Author:

The paper of Hosaka, K et al described the major role of FGF2 in mediating resistance to treatments targeting VEGF and PDGF. Very interestingly, the combination of anti VEGF and anti-PDGFR bypassed the resistance mediated by FGF2 overexpression. This breakthrough is breast cancers and sarcoma. The associated molecular mechanism involved in the aggressiveness of FGF2+ tumors depends on the recruitment of pericytes on tumor micro-vessels that can be eradicated by anti-PDGFR. Destruction of pericytes exposed tumor vessels to anti-VEGF therapy. This very important discovery will serve to stratify patients that will benefit of the anti-VEGF/anti-PDGF combo.

Major questions

- 1- What is the proliferation rate of cells overexpressing FGF2? It is also possible that FGF2 overexpressing cells also overproduced VEGF and PDGF and that the dose of anti VEGF and imatinib became limiting? Hence, PDGF and VEGF levels should be evaluated in FGF2- and FGF2+ cell supernatant.
- 2- Artefactually, anti VEGF or imatinib may decrease the production of FGF2 by the transduced cells by inhibiting the promoter of the expression vector. This point needs to be discussed.
- 3- The effect of anti-VEGF and anti-PDGFR should be evaluated directly on tumor cells in vitro to dissociate the effect on tumor cells from those on the microenvironment
- 4- According to the paradigm of R Jain, inhibition of abnormal blood vessels should improve perfusion which was not the case in Figures 2 and 4. This point needs to be discussed.
- 5- Why in sarcoma, imatinib decreased CD31 in FGF2- and FGF2+ tumors (Sup2) and had no effect on tumor growth in FGF2+ tumors? In the discussion it is said that FGFR inhibition did not impact tumor growth. It deserves to be included as a supplementary figure.

6- What is the relationship with human tumor? Does FGF2 overexpression is linked to shorter survival in breast cancers or in sarcoma or other cancer types? TCGA analysis would add a translational part to this excellent article.

Minor questions

Lane 7 page 5: 392 ng/g: What is the comparison with normal mammary epithelia cells

Lane 17 page 5: Imatinib was primarily designed to target BCR/ABL in chronic myeloid leukemia and is also used to treat the GIST

Lane 4 page 6: antiangiogenic resistant

Lane 12 page 8: To demonstrate synergy specific mathematic analysis should be performed. Hence, synergistic should be removed. Id lane 13

Lane 19/20 page 8: ".....demonstrating that vascular suppression is the principal mechanism of antitumor activity". The double treatment seems to also affect tumor cell proliferation? Which cells undergo apoptosis ? Tumor cells endothelial cells or both ?

According to the paradigm of R Jain, inhibition of abnormal blood vessels should improve perfusion which was not the case in Figures 2 and 4 after double treatments (pimonidazole). This point needs to be discussed.

Lane 3 page 10: "rather than suppression" to be corrected
Why FGF2+ tumors are more hypoxic?

Reviewer #3:

Remarks to the Author:

Y. Cao and colleagues report - based on mouse studies only - that antiangiogenic tumor therapy could be improved by combining anti-VEGF with anti-PDGF treatment. Mechanistically they claim that this occurs through targeting pericytes via anti-PDGF and subsequently by targeting the more immature vessels with anti-VEGF. It is well known that anti-VEGF therapy has many limitations and this might be resolved by using combinatory treatment approaches (e.g. combining anti-VEGF with anti-Angpt2). It is also well known that anti-VEGF works best on immature capillaries with poor pericyte coverage. As such, the anti-PDGF approach is interesting as it is well known that targeting the PDGF axis impairs vessel coverage with pericytes. This manuscript also claims that tumor produced FGF2 is a resistance mechanism in anti-VEGF or anti-PDGFR monotherapy. While FGF2 can still not be sufficiently targeted, the anti-PDGF + anti-VEGF approach could somehow help to resolve this. This is a very strong finding that could be easily followed up in human studies. Nevertheless the manuscript still appears somehow preliminary and the mechanistic insights are not yet fully clear.

Major points

- 1) Antiangiogenic therapy in the clinical setting is usually applied together with classical chemotherapy ("vascular normalization"). The authors should test whether anti-VEGF+anti-PDGFRb improves the outcome of chemotherapy in their mouse models.
- 2) The combined treatment increases tumor hypoxia and apoptosis. The images shown do not show if there is also necrosis in parts of the tumors.
- 3) Does the combined anti-angiogenic treatment increase the metastasis rate? As the treatment strongly increases tumor hypoxia it could lead to selection of highly aggressive tumor cells.
- 4) The analysis of the immune cells in the tumors is superficial (only Iba+ cells). A much more thorough analysis of other immune cells in particular also CD4 and CD8 T cells would be required. Ideally the IHC approach would be combined with cell sorting to determine the degree of the various immune cells in the tumors.
- 5) It is surprising that the authors did not check how their combined treatment acts on the normal vasculature in different organs (in particular kidney and endocrine organs). Does this alter vessel density, permeability, leakiness, pericyte coverage? Does the combined treatment impair kidney function (e.g. test for proteinuria and or blood pressure)? In addition, there is no data shown how

the mice tolerate these treatment (at least body weight curves would be needed).

6) It is also surprising to see that the authors did not follow up how FGF2 – released from the tumor cells – acts on endothelial cells and pericytes in the tumor microenvironment or at least in more simplified co-culture models. Just citing some literature reports about FGF2-induced changes in PDGF signaling components appears not to be sufficient.

Others

All of the figures are fully packed and the resolution of the images need to be approved. In the current form they are all difficult to view.

Our point-by-point responses to the reviewers' comments are as follows:

Reviewers' comments:

Reviewer #1 (Remarks to the Author):

Comments: This paper is of broad interest to the readership of Nature Communications for the following reasons

- 1) It identifies FGF2 as a potential biomarker of Avastin resistant tumors
- 2) It identifies FGF2 as a mechanism of resistance
- 3) It demonstrates that the combination of VEGF blockade and imatinib are effective against FGF positive tumors, even when the monotherapy of each one is ineffective. This should lead to reconsideration that if a drug is ineffective in a given clinical setting, it may contribute to the effectiveness of a combination therapy. This may lead to a general re-evaluation of how we determine drug efficacy in clinical trials
- 4) Imatinib inhibits FGF signaling despite not inhibiting FGF receptor signaling.

Response: We thank the reviewer for these positive comments and accurately understanding of our findings. These comments are very encouraging for us to continue research work along this line.

Specific comments: Imatinib monotherapy appears to induce a hypervascular and hyperpermeable phenotype compared with other treatments. Please comment on this. One potential mechanism might be the induction of ANgpt4, and Angpt4 blockade with propranolol might ameliorate this phenomenon- see Propranolol exhibits activity against hemangiomas independent of beta blockade.

NPJ Precis Oncol. 2019 Nov 1;3:27.

Response: Completely agree. We believe that ablation of pericytes from tumor vessels by imatinib resulting in hypervascularization and increased vascular permeability by the following possible mechanisms: 1) Detaching pericytes allows endothelial cells being exposed to other angiogenic stimuli such as VEGF. The increased exposure of endothelial cells to VEGF would increase vascular sprouting and thus increased microvascular density was observed in the imatinib-treated tumors; 2) Shedding pericytes by imatinib would increase vascular leakiness; 3) As the reviewer suggested, Angpt4 might be involved in these processes.

Comments: 2) Please reference the following articles on PDGF signaling (Cooperative benefit for the combination of rapamycin and imatinib in tuberous sclerosis complex neoplasia.

Vasc Cell. 2012 Jul 5;4(1):11. doi)

(Malignant transformation of human cells by constitutive expression of platelet-derived growth factor-BB.

J Biol Chem. 2005 Apr 8;280(14):13936-43.)

Response: In the revised manuscript, we have cited these publications.

Reviewer #2 (Remarks to the Author):

Comments: The paper of Hosaka, K et al described the major role of FGF2 in mediating resistance to treatments targeting VEGF and PDGF. Very interestingly, the combination of anti-VEGF and anti-PDGFR bypassed the resistance mediated by FGF2 overexpression. This breakthrough is breast cancers and sarcoma. The associated molecular mechanism involved in the aggressiveness of FGF2+ tumors depends on the recruitment of pericytes on tumor micro-vessels that can be eradicated by anti-PDGFR. Destruction of pericytes exposed tumor vessels to anti-VEGF therapy.

This very important discovery will serve to stratify patients that will benefit of the anti-VEGF/anti-PDGF combo.

Response: We thank the reviewer for his/her positive comments and for considering our work being interesting. We also thank the reviewer for accurately understanding our work and indicating the clinical significance for antiangiogenic combination therapy.

Major questions

Comment: 1- What is the proliferation rate of cells overexpressing FGF2? It is also possible that FGF2 overexpressing cells also overproduced VEGF and PDGF and that the dose of anti VEGF and imatinib became limiting? Hence, PDGF and VEGF levels should be evaluated in FGF2- and FGF2+ cell supernatant.

Response: We thank the reviewer for these important comments. In order to address the reviewer's comments, we have performed new experiments. We show that in both breast cancer and fibrosarcoma cells, overexpression of FGF2 did not alter cell proliferation rates. We have included these new data in the revised manuscript (Fig. S1). We have analyzed the production of VEGF and PDGF-B in FGF2-overexpressing and control cells and we see no differences of VEGF and PDGF-B production in FGF2-overexpressing and control cells. We have included these new data in the revised manuscript (Fig. S1).

Comment: 2- Artefactually, anti VEGF or imatinib may decrease the production of FGF2 by the transduced cells by inhibiting the promoter of the expression vector. This point needs to be discussed.

Response: Agree. We have discussed this interesting point in the Discussion section of the manuscript.

Comment: 3- The effect of anti-VEGF and anti-PDGFR should be evaluated directly on tumor cells in vitro to dissociate the effect on tumors cells from those on the microenvironment

Response: Completely agree. We have performed new experiments using anti-VEGF and anti-PDGFR β for treating breast cancer and fibrosarcoma cells. Anti-VEGF had no impacts on proliferation of both cancer cell types. While anti-PDGFR β had no effect on fibrosarcoma cells, it slightly inhibited breast cancer cell proliferation. However, this inhibition is indistinguishable between FGF2-overexpressing and control tumor cells. We have included these data in the revised manuscript (Fig. S1).

Comment: 4- According to the paradigm of R Jain, inhibition of abnormal blood vessels should improve perfusion which was not the case in Figures 2 and 4. This point needs to be discussed.

Response: Completely agree. In the revised manuscript, we have discussed this point.

Comment: 5- Why in sarcoma, imatinib decreased CD31 in FGF2- and FGF2+ tumors (Sup2) and had no effect on tumor growth in FGF2+ tumors? In the discussion it is said that FGFR inhibition did not impact tumor growth. It deserves to be included as a supplementary figure.

Response: In Figure S2 of the original manuscript, we showed that imatinib monotherapy had no significant effect on suppression of CD31+ vessels in FGF2-overexpression tumors (Fig. S2c of the original manuscript). We only see inhibition of FGF2- tumor by imatinib. We have included these data in the revised manuscript (Fig. S4c in the revised manuscript). In the revised manuscript, we have included the FGFR inhibition data in Fig. S10 of the revised manuscript.

Comment: 6- What is the relationship with human tumor? Does FGF2 overexpression is linked to shorter survival in breast cancers or in sarcoma or other cancer types? TCGA analysis would add a translational part to this excellent article.

Response: We thank the reviewer for raising this important question. Based on the reviewer's suggestion, we have correlated FGF2 expression with survival in several tumor types, including breast, ovarian, bladder, and pancreatic cancers. Our data show that FGF2 expression was correlated with survival disadvantages. We have included these TCGA data in the revised manuscript (Fig. 7).

Minor questions

Comment: Lane 7 page 5: 392 ng/g: What is the comparison with normal mammary epithelia cells

Response: We have measured the FGF2 expression in normal mammary epithelial cells and the concentration was 70.034 pg/ml.

Comment: Lane 17 page 5: Imatinib was primarily designed to target BCR/ABL in chronic myeloid leukemia and is also used to treat the GIST

Response: We have included this information of imatinib in the revised manuscript.

Comment: Lane 4 page 6: antiangiogenic resistant

Response: We thank the reviewer for finding this typo error. We have corrected in the revised manuscript.

Comment: Lane 12 page 8: To demonstrate synergy specific mathematic analysis should be performed. Hence, synergistic should be removed. Id lane 13

Response: We have removed synergistic in the revised manuscript.

Comment: Lane 19/20 page 8: “.....demonstrating that vascular suppression is the principal mechanism of antitumor activity”. The double treatment seems to also affect tumor cell proliferation? Which cells undergo apoptosis ? Tumor cells endothelial cells or both ?

Response: Agree. We think that vascular suppression will affect both tumor cell proliferation and apoptosis. As seen in many other tumor models, targeting tumor angiogenesis would certainly affect tumor cell proliferation and apoptosis. In this study, we show that combination therapy inhibits proliferation of CD31+ cells and also inhibits tumor cell proliferation. We also observed an increased cellular apoptosis in double drug-treated tumors. The apoptotic cells are mainly restricted to tumor cells, but not endothelial cells. These data are now presented in Fig. 3, Fig. 5, and Fig. 6. We hope that this explanation makes sense for the reviewer.

Comment: According to the paradigm of R Jain, inhibition of abnormal blood vessels should improve perfusion which was not the case in Figures 2 and 4 after double treatments (pimonidazole). This point needs to be discussed.

Response: In our hands, we have not seen increased blood perfusion in anti-VEGF and imatinib combination-treated tumors. In contrast, we double drug-treated tumors exhibited a higher degree of hypoxia. We have discussed this point in the revised manuscript.

Comment: Lane 3 page 10: “rather than suppression” to be corrected Why FGF2+ tumors are more hypoxic?

Response: We removed “rather than suppression” in the revised manuscript. It is likely that the accelerated growth rate might be the reason for causing higher hypoxia.

Reviewer #3 (Remarks to the Author):

Comment: Y. Cao and colleagues report - based on mouse studies only - that antiangiogenic tumor therapy could be improved by combining anti-VEGF with anti-PDGF treatment. Mechanistically they claim that this occurs through targeting pericytes via anti-PDGF and subsequently by targeting the more immature vessels with anti-VEGF. It is well known that anti-VEGF therapy has many limitations and this might be resolved by using combinatory treatment approaches (e.g. combining anti-VEGF with anti-Angpt2). It is also well known that anti-VEGF works best on immature capillaries with poor pericyte coverage. As such, the anti-PDGF approach is interesting as it is well known that targeting the PDGF

axis impairs vessel coverage with pericytes. This manuscript also claims that tumor produced FGF2 is a resistance mechanism in anti-VEGF or anti-PDGFR monotherapy. While FGF2 can still not be sufficiently targeted, the anti-PDGF + anti-VEGF approach could somehow help to resolve this. This is a very strong finding that could be easily followed up in human studies. Nevertheless the manuscript still appears somehow preliminary and the mechanistic insights are not yet fully clear.

Response: We thank the reviewer for these insightful and authoritative comments, which highlight the importance of combination therapy in improving clinical outcomes. Indeed, FGF2 is not sufficiently targeted and our findings would be easily followed up by human studies.

Major points

Comment: 1) Antiangiogenic therapy in the clinical setting is usually applied together with classical chemotherapy (“vascular normalization”). The authors should test whether anti-VEGF+anti-PDGFRb improves the outcome of chemotherapy in their mouse models.

Response: We thank the reviewer for this excellent suggestion. Based on the reviewer’s comments, we have performed new experiments by combining chemotherapy. We used 5FU at a high (60 mg/kg) and low (10 mg/kg) once/week for monotherapy and combination therapy. At 10 mg/kg, 5FU had impact on tumor growth in E0771 breast cancer. Combination of 5FU + anti-VEGF + imatinib did not further increase the antitumor activity. At 60 mg/kg, 5FU alone displayed a potent antitumor activity and triple combination 5FU + anti-VEGF + imatinib did produce any additive effects. In fibrosarcoma, 10 mg/kg of 5FU produced a potent antitumor activity and triple combination 5FU + anti-VEGF + imatinib did not increase antitumor activity of Imatinib + anti-VEGF. At 60 mg/kg, anti-VEGF + imatinib slightly, but statistically significant, increased the antitumor activity of 5FU. We have included these new results in Fig. S8 of the revised manuscript.

Comment: 2) The combined treatment increases tumor hypoxia and apoptosis. The images shown do not show if there is also necrosis in parts of the tumors.

Response: We have quantified the necrotic areas of treated and non-treated tumors. In FGF2⁻ control breast and sarcoma tumors, combination therapy increased the necrotic areas. However, combination therapy did not increase necrosis in FGF2⁺ tumors. These findings suggest that apoptosis is primary process responsible for tumor cell death. We have included these data are now included in the revised manuscript (Fig. S2 and Fig. S5).

Comment: 3) Does the combined anti-angiogenic treatment increase the metastasis rate? As the treatment strongly increases tumor hypoxia it could lead to selection of highly aggressive tumors cells.

Response: We thank the reviewer for this excellent question. The reviewer’s point is well-taken. However, we focus our present study on drug responses. We plan to do thorough metastatic studies in the future and prefer to publish those findings separately. We thank the reviewer for his/her understanding.

Comment: 4) The analysis of the immune cells in the tumors is superficial (only Iba+ cells). A much more thorough analysis of other immune cells in particular also CD4 and CD8 T cells would be required. Ideally the IHC approach would be combined with cell sorting to determine the degree of the various immune cells in the tumors.

Response: We thank the reviewer for this excellent comment. We have performed new experiments by employing IHC and FACS analyses. FACS analysis showed that FGF2⁺ tumors contain an increased F4/80 population, which is consistent with the IHC results. Interestingly, the total CD3⁺T-cell subpopulation was significantly decreased in FGF2⁺ E0771 tumors relative to FGF2⁻ control E0771 tumors. Combination therapy did not alter the ratios. In the CD4⁺ population, there was a significant decrease in FGF2⁺ tumors relative to control tumors. Again, combination therapy did not alter the ratios. In CD8⁺ subpopulation, there were no significant differences between FGF2⁻ control and FGF2⁺ tumors. IHC findings reconciled with FACS data. These new data were included in the revised manuscript (Fig. S3).

In the fibrosarcoma model, except macrophages the T-total population, CD4⁺ subpopulation, and CD8⁺ subpopulation remained unchanged between FGF2⁻ and FGF2⁺ tumors and all treated groups. These new findings are now included in the revised manuscript (Fig. S6).

Comment: 5) It is surprising that the authors did not check how their combined treatment acts on the normal vasculature in different organs (in particular kidney and endocrine organs). Does this alter vessel density, permeability, leakiness, pericyte coverage? Does the combined treatment impair kidney function (e.g test for proteinuria and or blood pressure)? In addition, there is no data shown how the mice tolerate these treatment (at least body weight curves would be needed).

Response: We apologized for not presenting the impact of combination therapy on alterations of normal vasculatures in different organs, including the kidney and endocrine organs. We should emphasize that we used a relatively low dose (2.5 mg/kg) of an anti-VEGF neutralizing antibody for our combination studies. It appeared that anti-VEGF monotherapy and anti-VEGF + imatinib combination significantly reduced microvessel density in thyroid. However, we did not see significant reduction of microvessels in other organs, including kidney and skeletal muscles. Consistent with reduction of microvessel density in the combination-treated thyroid, blood perfusion measured by 2000-kD dextran was accordingly decreased. However, pericyte and vascular leakiness (extravasation of 70-kD dextran) remained unchanged in the combination treated groups relative to the controls.

Additionally, systemic treatment of mice with the combination therapy did not affect the body weight and proteinuria. However, slightly increases of systolic and diastolic blood pressures were observed in the combination therapy-treated animals. These new experimental data are now included in the revised manuscript (Fig. S9).

Comment: 6) It is also surprising to see that the authors did not follow up how FGF2 – released from the tumor cells – acts on endothelial cells and pericytes in the tumor microenvironment or at least in more simplified co-culture models. Just citing some literature

reports about FGF2-induced changes in PDGF signaling components appears not to be sufficient.

Response: We thank the reviewer for raising this important issue. To detect biological functions of FGF2 released from tumor cells on endothelial cells and pericytes, we collected conditioned media from FGF2⁺ and control tumor cells. These conditioned media were used to stimulate endothelial cell and pericyte proliferation. In this experimental setting, we have used recombinant FGF2 as a control. As shown in Figure S1 i and j, conditioned media from FGF2⁺ breast cancer and fibrosarcoma significantly stimulated endothelial cell and pericyte proliferation, which is comparable to that induced by exogenously added recombinant FGF2. There are statistically significant differences between FGF2⁺ tumor cell conditioned media compared to those from the FGF⁻ tumor cells. These new data demonstrated that FGF2 released from tumor cells are biologically active. We have included these new data in the revised manuscript (Fig. S1).

Comment: Others

All of the figures are fully packed and the resolution of the images need to be approved. In the current form they are all difficult to view.

Response: Completely agree. We have provided high-resolution images in the revised manuscript. We apologize for not providing the low-resolution images in the original manuscript. For publication, we will certainly provide the original images that are in high quality.

Reviewers' Comments:

Reviewer #1:

Remarks to the Author:

The authors have satisfied my concerns. I recommend acceptance

Reviewer #2:

Remarks to the Author:

The authors have correctly answer to all the cricisms raised during the first round of reviewing. The paper is excellent and the previous criticisms were to improve the quality of the manuscript. I totally agree to publish this original paper in Nat Comm

Reviewer #3:

Remarks to the Author:

The authors have addressed all of my concerns. The manuscript has substantially been approved and appears to be ready for publication.

Andreas Fischer